# The gag-like gene *RTL8* antagonizes PEG10-mediated virus like particles

**Will Campodonico**[1], **Harihar M. Mohan**[2,3], **Phuoc T. Huynh**[1,4], **Holly H. Black**[1], **Cristina I. Lau**[1], **Henry L. Paulson**[2], **Lisa M. Sharkey**[2], **Alexandra M. Whiteley**[1]*

1 Department of Biochemistry, University of Colorado, Boulder, CO, United States of America, 2 Department of Neurology, University of Michigan Medical School, Ann Arbor, MI, United States of America, 3 Cellular and Molecular Biology Program, University of Michigan Medical School, Ann Arbor, MI, United States of America, 4 Molecular, Cellular and Developmental Biology Program, University of Colorado, Boulder, CO, United States of America

* alexandra.whiteley@colorado.edu

**Data Availability Statement:** All relevant data are within the paper and its Supporting Information files.

**Funding:** National Institute of Neurological Disorders and Stroke, R35NS122302, Henry L.

## Abstract

*PEG10* is a retroelement-derived *Mart*-family gene that is necessary for placentation and has been implicated in neurological disease. PEG10 resembles both retrotransposon and retroviral proteins and forms virus-like particles (VLPs) that can be purified using iodixanol ultracentrifugation. It is hypothesized that formation of VLPs is crucial to the biological roles of PEG10 in reproduction and neurological health. Here, we describe the regulation of PEG10 VLP formation and release in human cells with a role for the related *Mart* gene *RTL8*. RTL8 resembles a truncated form of PEG10 that shares homology with the N-terminal gag-like capsid domain. Alone, RTL8 is unable to form VLPs, but was incorporated into PEG10-derived particles. RTL8 co-expression decreased the abundance of PEG10 VLPs and increased intracellular levels of PEG10, suggesting a model where RTL8 inhibits PEG10 VLP formation or release. Consistent with this model, RTL8 bound to the N-terminal domain of PEG10 capsid, and modulation of RTL8 influenced PEG10-derived VLP abundance in naturally producing cells. RTL8 is broadly expressed in many of the same tissues as PEG10, including in human brain. Taken together, these results describe a novel antagonistic relationship between two human retroelement-derived genes and have implications for our understanding of PEG10 biology and disease.

## Introduction

Mammalian genomes contain hundreds of genes derived from transposable elements (TEs), a subset of which have been demonstrated to play adaptive roles in the cell [1–6]. These include genes from LTR retrotransposons and retroviruses, which share a classic gag-pol (or gag-pol-env) genetic structure [7–12]. In the TE lifecycle, the gag protein encodes domains necessary for capsid formation, while the longer gag-pol form also contains enzymatic domains necessary for replication. There are almost 100 gag-like genes in the human genome, many of which encode partial or truncated gag domains with poorly understood function [8, 12, 13]. For

Paulson, Harihar M. Mohan, and Lisa M. Sharkey University of Michigan, Rackham Predoctoral Fellowship, Harihar M. Mohan University of Colorado Venture Partners, Will Campodonico, Dr. Alexandra M. Whiteley National Institute of Neurological Disorders and Stroke, R01NS131660, Dr Alexandra M. Whiteley.

**Competing interests:** W.C. and A.M.W. are authors on a patent related to the work presented in this manuscript. This does not alter our adherence to PLoS One policies on sharing data and materials.

some gag-like genes, capsid formation and particle release still occurs *in vitro* and in cells [5, 14–17].

In some mammals, gag-like genes have been found that interfere with the capsid formation, release, or breakdown of infectious elements [18–22], thereby protecting the organism against disease. No human gag-like genes have been identified that share a similar antiviral activity, though an exapted *env*-like gene was recently identified that protects against retrovirus infection [11]. Instead, human gag-like genes play roles in gene expression [23] and have been implicated in neuronal function and reproduction. For example, the gag-like gene *Arc* is necessary for AMPA receptor function [6, 24] and long-term potentiation [25]. The gag-like gene *Paternally Expressed Gene 10* (*PEG10*), which uniquely contains both *gag* and *pol* open reading frames, and can generate both gag and gag-pol fusion proteins, is necessary for placentation in mice [4, 5, 26]. However, high levels of PEG10 have also been linked to cancer and neurological disease in humans [17, 27, 28], indicating that it may play a pathological role when dysregulated.

Formation of a capsid and release of a virus-like particle (VLP) made of Arc protein appears to be central to its adaptive role in the organism [14, 15]. PEG10 can also form VLPs that are released from human cells [5, 16, 17], though it is unknown whether VLP formation contributes to its beneficial or harmful activities. Because of their ability to form VLPs, human gag-like genes like Arc and PEG10 are attractive in the bioengineering field for the development and production of biocompatible vaccine-delivery mechanisms. As a human gene product, these proteins are not thought to be immunogenic. Further, thanks to the presence of a CCHC-type zinc finger, PEG10 can naturally encapsulate mRNA [5, 16]. Therefore, an understanding of the cellular processes that govern VLP formation is relevant for an understanding of PEG10's biological function as well as its potential utility as a tool for vaccine development.

Here, we describe a novel role for the human gag-like gene *RTL8* in the restriction of *PEG10*-derived VLP release. *RTL8* is a gag-like gene from the same Mart family as PEG10 and encodes a 113 AA protein with a truncated capsid domain and no zinc finger [29]. RTL8 is found in PEG10 VLPs, binds to the PEG10 N-terminal lobe, and is associated with increased intracellular PEG10 levels. Further, cell lines with high RTL8 expression tend to release fewer PEG10-derived VLPs. RTL8's antagonism of PEG10 appears to be species-specific, suggesting that the two genes may have co-evolved in conflict. Together, these findings suggest a model in which RTL8 specifically incorporates into PEG10 VLPs, thereby inhibiting their assembly or release. These results illuminate a new role for *RTL8* in the inhibition of PEG10 activity and may have important implications for the biology of PEG10, as well as the production and utility of PEG10-derived VLPs.

## Materials and methods

### Constructs

See Table 1.

### Cloning

All cloning was performed by Gibson assembly (Gibson HiFi master mix, Invitrogen) and transformed into chemically competent DH5α *E. coli* cells (Invitrogen). All constructs used in the study are listed in Table 1. Transformed *E. coli* were plated on either 50 mg/mL kanamycin (Teknova) or 100 mg/mL carbenicillin (Gold Biotechnology) LB agar (Teknova) plates overnight at 37˚C. Single colonies were picked and grown overnight in 5 mL LB Broth (Alfa Aesar) with kanamycin or carbenicillin at 37˚C with shaking at 220 rpm. Shaking cultures were mini-prepped (Zymo) and sent for Sanger Sequencing (Azenta). Sequence verified samples were

**Table 1. Constructs used in this study.**

| Construct Name | Insert | Tag | Species | NCBI protein reference | Vector |
|---|---|---|---|---|---|
| eGFP | eGFP | 2x HA (N-terminal) | *Aequorea victoria* | 6XZF_A | pcDNA3.1 |
| gag-pol | PEG10 (AA1-708) | 2x HA (N-terminal) | *Homo sapiens* | NP_055883.2 | pcDNA3.1 |
| gag | PEG10 (AA1-325) | 2x HA (N-terminal) | *Homo sapiens* | NP_001035242.1 | pcDNA3.1 |
| capsid$^{NTD}$ | PEG10 (AA1-160) | 2x HA (N-terminal) | *Homo sapiens* | | pcDNA3.1 |
| capsid$^{CTD}$ | PEG10 (AA161-259) | 1x HA (C-terminal) | *Homo sapiens* | | pcDNA3.1 |
| N-terminal fragment | PEG10 (AA1-111) | 2x HA (N-terminal) | *Homo sapiens* | | pcDNA3.1 |
| C-terminal fragment | PEG10 (AA112-325) | 2x HA (C-terminal) | *Homo sapiens* | | pcDNA3.1 |
| FLAG-*Hs*RTL8a-3'UTR | RTL8a (AA1-113), with 700bp 3'UTR | 3x FLAG (N-terminal) | *Homo sapiens* | NP_001071640.1 | pcDNA3.1 |
| FLAG-*Hs*RTL8b-3'UTR | RTL8b (AA1-113), with 859bp 3'UTR | 3x FLAG (N-terminal) | *Homo sapiens* | NP_001071641.1 | pcDNA3.1 |
| FLAG-*Hs*RTL8c-3'UTR | RTL8c (AA1-113), with 821bp 3'UTR | 3x FLAG (N-terminal) | *Homo sapiens* | NP_001071639.1 | pcDNA3.1 |
| HA-*Hs*RTL8c-3'UTR | RTL8c (AA1-113), with 821bp 3'UTR | 2x HA (N-terminal) | *Homo sapiens* | NP_001071639.1 | pcDNA3.1 |
| *Mm*RTL8b | RTL8b (AA1-113) | 3x FLAG (N-terminal) | *Mus musculus* | NP_001018073.1 | pcDNA3.1 |
| gag-pol flow reporter (Human) | PEG10 (AA1-708) | Dendra2 (C-terminal), IRES-CFP | *Homo sapiens* | NP_055883.2 | pDendra2 |
| gag-pol$^{FFS}$ flow reporter (Human) (used for WB) | PEG10 (AA1-708) *954G>T/ 956A>C/ 961T+A/ 969C>G/ 972C>A/ 974C+AGT/ 975T>A/ 981G>A/ 987TTCA>CAGC/ 993G>A | Dendra2 (C-terminal), IRES-CFP | *Homo sapiens* | NP_055883.2 | pDendra2 |
| gag-pol flow reporter (Mouse) | PEG10 (AA1-1005) | Dendra2 (C-terminal), IRES-CFP | *Mus musculus* | NP_570947.2 | pDendra2 |

then grown in 50 mL LB Broth overnight with appropriate antibiotic at 37˚C with shaking at 220 rpm. 50 mL cultures were midi-prepped (Zymo) for transfection.

## Cell lines

hTR-8/SVneo (CRL-3271), SK-N-SH (HTB-11), T98G (CRL-1690), CCF-STTG1 (CRL-1718), BE(2)-M17 (CRL-2267), and M059K (CRL-2365) cells were purchased from ATCC. A549 and U-87 MG cells were a gift from Dr. Roy Parker (Department of Biochemistry, CU Boulder). HepG2 cells were obtained from ATCC (HB-8065) via the CU Boulder Biochemistry Shared Cell Culture Facility. HEK293 cells were a gift from Dr. Ramanujan Hegde (Medical Research Council Laboratory of Molecular Biology, Cambridge England).

Induced cortical neurons were differentiated and cultured as previously described [30]. Cells were cultured on Matrigel (Corning) coated plates. Following differentiation, cells were maintained with half-media changes as described. The removed media from these half-media changes was collected, flash frozen, and stored at -80˚C. Once ~20mL media had been collected, samples were thawed and virus-like particles isolated as described below.

All cells were maintained at 37°C with 5% $CO_2$. HEK293 and A549 cells were maintained in DMEM (Invitrogen) supplemented with 1% penicillin/streptomycin (Invitrogen), 1% L-glutamine (R&D Systems, Inc.), and 10% FBS (Millipore Sigma). hTR-8/SVneo cells were maintained in RPMI 1640 (Invitrogen) supplemented with penicillin/streptomycin, L-glutamine, and 10% FBS. CCF-STTG1 was cultured in RPMI-1640 containing 20% FBS and penicillin/streptomycin. HepG2 and U87 MG cells were maintained in MEM (Invitrogen) supplemented with penicillin/streptomycin, L-glutamine, and 10% FBS. SK-N-SH and T98G were cultured in MEM/EBSS (Cytiva Life Sciences) containing 1 mM sodium pyruvate (Invitrogen), 0.1 mM NEAA (Invitrogen), 10% FBS and penicillin/streptomycin. BE(2)-M17 was cultured in DMEM/F12 (Invitrogen) containing 10% FBS and penicillin/streptomycin. M059K was cultured in DMEM/F12 containing 0.05 mM NEAA, 20% FBS and penicillin/streptomycin.

## Transfection

Cells were grown to 70% confluency and transfected with Lipofectamine 2000 (ThermoFisher) according to manufacturer's instructions. For 6-well plates, 2.5µg plasmid DNA was transfected per well. For 12-well plates, 1µg plasmid DNA was transfected per well. For 96-well plates, 0.1µg plasmid DNA was transfected per well. For cotransfections, equal mass amounts of each plasmid were added to the total amount listed above. Transfection mixture was prepared at a ratio of 1µg DNA:2.5µL Lipofectamine 2000. For siRNA, cells were transfected with 0.025 nmol of an RNA duplex containing the sequence 5'-GCUCCUACAUGUUCGUGGA-3' of human RTL8 from Horizon Discovery/Dharmacon. Unless otherwise stated, cells and media were harvested 48h following transfection.

## Virus-like particle isolation

Crude preparation: For endogenous VLP production, T75 flasks were plated at 70% confluency. For overexpression experiments, cells were plated at 70% confluency in 6-well plates and were transfected as described above. Cells were grown for 48 hr and conditioned media was collected. Media was first centrifuged at 2700 x g for 10 min to remove cellular debris. The supernatant was harvested, then spun by ultracentrifugation using a preparative ultracentrifuge (Beckman Coulter) at 134,000 x g (Beckman SW41Ti rotor) for 4 hr at 4°C over a 30% sucrose (MP Biomedicals) cushion. Media and sucrose were aspirated and the VLP-containing pellet was resuspended in lysis buffer for western blot.

Iodixanol preparation: Iodixanol (Optiprep, Sigma Aldrich) gradients were prepared using PBS-MK buffer to generate fractions of 60%, 40%, and 25% iodixanol in PBS-MK, and 15% iodixanol in PBS-MK with 1M NaCl. The 60% and 25% fractions were pre-mixed with phenol red to color the gradient. Iodixanol steps were layered in Beckman Coulter 38.5 mL open-top, thin-wall ultra-clear tubes using a stripette. 5 mL of the 60% step, 5 mL of the 40% step, 6 mL of the 25% step, and 8 mL of the 15% step were sequentially layered. Conditioned medium was spun at 2,700g for 15 minutes at 4°C to remove cell debris and up to 8 mL of the supernatant was layered on top of iodixanol. Then, PBS was added to balance the tubes in an SW32 Ti rotor. Media was spun at 20,000 rpm for 18 hours at 4°C.

After spinning, parafilm was put over the top of the tube to seal and an 18 gauge needle (BD) was used to puncture just below the 60%-40% interface. Drops were collected into microcentrifuge tubes in approximately 1 mL fractions, starting with #1. Color change and change in flow rate was used to denote the approximate locations of iodixanol steps across collected fractions.

Protein was isolated from fractions using methanol:chloroform precipitation and 500 µL of proteinaceous starting material. After precipitation, protein was resolubilized in 12µL of 8M

urea buffer to which 3μL of 5x Laemmli sample buffer was added. The entire sample was run on a 4–12% Bis-tris gel and western blotted according to methods.

## Negative stain electron microscopy

A small volume (6–8μl) of sample from iodixanol centrifugation was applied to a carbon-coated, copper mesh TEM grid for 1–2 minutes, washed twice with water, and then negative stained with 2% aqueous uranyl acetate. TEM imaging was done on a FEI/TFS Tecnai T12 Spirit TEM, operating at 100 kV, with an AMT CCD. At least 20 images were collected from each sample at approximately 0.4 pixels/nm of different fields of view for quantitation of VLPs.

## Human brain tissue

Fresh frozen post-mortem tissue samples from the frontal cortex and substantia nigra of age-matched non-demented (control, n = 3) patients were obtained from the University of Michigan Brain Bank (University of Michigan, Ann Arbor, MI) and used for analysis in Fig 5. Brain tissue was collected with patient consent, and protocols were approved by the Institutional Review Board of the University of Michigan and abide by the Declaration of Helsinki principles.

Soluble protein lysate was obtained as described previously [31]. The tissue was suspended in 10 volumes of Brain Homogenization Buffer (50 mM Tris, pH 7.4, 274 mM NaCl, 5 mM KCl with pepstatin, leupeptin, bestatin, and aprotinin protease inhibitors at 10 μg/mL and PMSF at 1 mM). The tissue was homogenized with a Tissue Tearor Model 985–370 on setting #3 for three times 10 s then centrifuged at 27,000 × g for 24 min at 4°C. The supernatant fraction was collected. A Bradford protein assay using Bio-Rad Protein Assay Dye Reagent Concentrate (Bio-Rad, Hercules, CA, #5000006) was performed on the lysates to determine total protein concentrations. The fractions were aliquoted and stored at -80°C until used in immunoblot assays. 40 μg of brain tissue homogenate was used for western blot.

## Western blotting

Cells were harvested by trypsinization, pelleted, and washed in PBS. For bulk cell lysate analysis, cell pellets were resuspended in in 8M urea (Fisher Chemical) containing 75 mM NaCl (Honeywell), 50 mM HEPES (Millipore Sigma) pH 8.5, with 1x cOmplete Mini EDTA-free protease inhibitor (Roche) and incubated at room temperature for 15 min. Lysate was cleared by centrifugation at 21,300 x g and the supernatant collected for western blot. Total protein was quantified by BCA assay (Pierce).

For endogenous protein analysis, 15μg total protein was loaded into NuPage 4–12% Bis-Tris precast protein gels (Life Technologies) per sample. For overexpression experiments, 4μg total protein was loaded per sample. Samples with equal protein content were generated by adding excess urea buffer to that all samples were of equivalent volume. Then, 5x Laemmli sample buffer supplemented with βME (Sigma Aldrich) was added to a final concentration of 1x. Proteins were separated by SDS-PAGE using 1x NuPage MES SDS running buffer (Life Technologies). For samples lysed in urea, wells of the protein gel were equilibrated in urea lysis buffer for 10 min, and urea washed out before loading protein samples.

Proteins were transferred to nitrocellulose membrane (Amersham) using the Invitrogen Mini Blot Module according to manufacturer's instructions. Membranes were blocked with Intercept (PBS) blocking buffer (LICOR) for 30 min at room temperature. Membranes were incubated with primary antibody overnight at 4°C. All antibodies used for western blot are listed in Table 2. Membranes were incubated with secondary antibodies for 30 min at room temperature. After both antibody incubations, membranes were washed three times for 5 min

**Table 2. Antibodies used in this study.**

| Target | Clonality | Species | Vendor | Catalog # | Dilution | Application |
|---|---|---|---|---|---|---|
| PEG10 | Polyclonal | Rabbit | Proteintech | 14412-1-AP | 1:1000 | WB |
| FAM127B (RTL8) | Polyclonal | Rabbit | Proteintech | 20282-1-AP | 1:1000 | WB |
| FLAG | Monoclonal M2 | Mouse | Sigma | F3165 | 1:5000 | WB, IP |
| FLAG | Monoclonal D6W5B | Rabbit | CST | 14793 | 1:5000 | WB |
| HA | Monoclonal HA-7 | Mouse | Sigma | H3663 | 1:5000 | WB, IP |
| HA | Monoclonal C29F4 | Rabbit | CST | 3724 | 1:5000 | WB |
| GAPDH | Monoclonal 14C10 | Rabbit | CST | 2118 | 1:5000 | WB |
| Tubulin | Monoclonal DM1A | Mouse | Novus | NB100-690 | 1:10000 | WB |
| ALIX | Monoclonal 3A9 | Mouse | CST | 2171S | 1:1000 | WB |
| IgG isotype control | Polyclonal | Rabbit | Invitrogen | 10500C | 1:600 | IP |
| α-mouse IgG 680 | Polyclonal | Goat | Licor | 926–68070 | 1:20000 | WB |
| α-mouse IgG 800 | Polyclonal | Goat | Licor | 926–32210 | 1:20000 | WB |
| α-rabbit IgG 680 | Polyclonal | Goat | Licor | 926–68071 | 1:20000 | WB |
| α-rabbit IgG 800 | Polyclonal | Goat | Licor | 926–32211 | 1:20000 | WB |
| α-mouse IgG HRP | Polyclonal | Goat | Jackson ImmunoResearch | 115-035-003 | 1:4000 | WB |
| α-rabbit IgG HRP | Polyclonal | Goat | Jackson ImmunoResearch | 111-035-003 | 1:4000 | WB |

each in TBST. Membranes were imaged on a LICOR Odyssey CLx and analyzed with LICOR ImageStudio software.

Unless otherwise stated, western blots were analyzed as follows. For cell lysate, target protein signal was normalized to tubulin as a loading control. This ratio was then normalized to the mean of the ratio for all samples within the experiment to account for technical variability across independent experiments. For endogenous VLP production, target protein signal was normalized to the mean target signal for all samples within the experiment. For VLP production from transfected PEG10, the overexpressed target protein signal from the VLP fraction was normalized to the tubulin-normalized cell lysate abundance of that target to account for variability in transfection efficiency. This ratio was then normalized to the mean of the ratio for all samples within the experiment to account for technical variation across independent experiments.

## Co-immunoprecipitations

For co-immunoprecipitation from HepG2 cells, cells were transfected with 5μg of empty vector or FLAG-eGFP was transfected 48 hr before harvesting cells. 5μg of rabbit isotype control or α-PEG10 antibody (Table 2) was coupled to Dynabeads Protein A by incubating with beads for 1 hr at 4°C with end-over-end rotation. Beads were washed twice with PBS and resuspended in 1mL of 5mM BS$^3$ for 30 min at room temperature with end-over-end rotation to crosslink antibodies. Excess crosslinker was quenched by adding 1M Tris HCl pH7.6 to a final concentration of 50 mM and incubating for 15 min at room temperature with end-over-end rotation. Beads were then washed thrice in 1x PBST at room temperature. Prior to IP, they were resuspended in IP lysis buffer.

Cells were trypsinized and washed once with warm PBS. They were lysed in IP lysis buffer containing 50mM Tris HCl pH7.6, 150mM NaCl, 1% NP40, 5% glycerol, 5mM EDTA and protease and phosphatase inhibitors for 30 min at 4°C with end-over-end rotation. Cell debris was cleared by centrifugation at 16,000 x g for 20 min at 4°C and the supernatant was collected for immunoprecipitation. Lysates were precleared by incubating with rabbit isotype control-conjugated beads for 1 hr at 4°C with end-over-end rotation. Precleared lysates were collected

and a BCA assay was performed to estimate protein concentration. 1mg and 10μg of lysate were used for IP and input respectively. Lysates were incubated with antibody-conjugated beads for 2 hr at 4˚C with end-over-end rotation. Beads were collected and washed thrice in IP lysis buffer. Bound proteins were eluted by boiling in 1x Laemmli buffer for 10 min and supernatants were used for western blots.

For crosslinking co-immunoprecipitation, cells were harvested by pipetting in ice cold PBS and centrifuged at 300 x g for 3 min, then resuspended in 300μL 0.1% PFA in PBS and incubated for 7 min at room temperature to crosslink proteins. Cells were collected by centrifugation at 300 x g for 3 min and washed three times with ice cold PBS. Crosslinked cells were lysed in 150μL lysis buffer containing 1% Triton X-100 (Sigma), 100mM NaCl (Honeywell), 10mM Hepes pH 7.5 (Sigma), 10mM EGTA (Sigma), 10mM EDTA (Sigma), and 1x protease inhibitor (Roche), and incubated for 30 min on ice. Lysate was pre-cleared by centrifugation at 16,000 x g for 5 min at 4˚C and the supernatant was collected for immunoprecipitation.

Beads were first crosslinked to antibodies. Primary antibodies (5μg) or isotype controls (5μL) were incubated with 40μL protein A (Invitrogen) or protein G (Pierce) bead slurry for 30 min to couple antibodies to beads. After coupling, beads were collected by centrifugation and the supernatant discarded. Unless otherwise stated, all incubations and washes were performed with end-over-end rotation at room temperature. All centrifugation to collect beads was performed 2,500 x g for 2 min. Beads were washed twice (1mL 0.2M sodium borate, pH 9 for 1 min), resuspended in 400μL 20mM dimethyl pimelimidate (DMP, Thermo) and incubated for 30 min to crosslink antibodies. After crosslinking, beads were washed twice (1mL 0.2M ethanolamine (EMD Millipore) pH 8 for 5 min) and incubated in 1.4mL 0.2M ethanolamine for two hr. Beads were then washed twice (1mL PBS for 5 min).

50μL crosslinked cell lysate and 50μL Triton lysis buffer were added to antibody-conjugated beads and incubated overnight at 4˚C with end-over-end rotation. After incubation, beads were collected by centrifugation and washed three times (500μL PBS, 0.1% Triton X-100 for 2 min). To elute proteins, beads were resuspended in elution buffer (20μL 5x Laemmli samle buffer and 30μL PBS) and incubated at 90˚C for 10 min. Following elution, the samples were centrifuged to collect the beads and the supernatant removed for analysis by western blot as described above. For SDS-PAGE, 7.5μL protein lysate (5%) and 25μL (50%) immunoprecipitated sample were loaded.

## Sequence alignment

Sequence alignment was performed on the EMBL-EBI MUSCLE online interface [32]. Alignments used the ClustalOmega algorithm [33] using the default parameters.

## Structure prediction

In S3b Fig in S1 File, structures of *Homo sapiens* PEG10 gag (AA 1–325) and RTL8 (AA 1–113) were modeled using Alphafold2 and threaded using DALI protein structure similarity server [34]. In S5b Fig in S1 File, PEG10 gag was modeled using the Phyre 2.0 web server [35] using the intensive modeling mode. All structures were visualized using UCSF Chimera [36].

Dimers of RTL8:PEG10 and PEG10:PEG10 were modeled using Alphafold2 with MMseqs2 at ColabFold [37] v1.5.2 found at:

https://colab.research.google.com/github/sokrypton/ColabFold/blob/main/AlphaFold2.ipynb#scrollTo=kOblAo-xetgx.

The entire protein sequence of RTL8c with the entire gag sequence (AA1-325) of PEG10 were used to model dimer structures of *Homo sapiens* RTL8 with *Homo sapiens* PEG10. Two sequences of PEG10 gag (AA1-325) were used to model PEG10 gag dimerization, and two

sequences of PEG10 CA$^{NTD}$ (AA1-154) were used to model NTD:NTD interactions. No template was used and all standard settings were selected. The resulting.pdb structures were visualized in Pymol. AA1-73 of PEG10, including the first alpha helix of the protein, were rendered invisible for all Figs. For RTL8:PEG10 visualizations, the CA$^{CTD}$ was rendered invisible.

### Flow cytometry

HEK293 cells were transfected in a 96-well or 48-well plate with plasmids encoding a PEG10--Dendra2 fusion protein followed by an IRES-CFP cassette under control of a CMV promoter. Cells were lifted by pipetting in FACS buffer (D-PBS, 2% FBS, 0.1% sodium azide) and analyzed on a FACSCelesta (BD Biosciences). At least 20,000 events were collected per sample on the cytometer. Flow cytometry analysis was performed using Flowjo software (Treestar). Single cells were first gated on FSC-A vs SSC-A, followed by gating of CFP+ cells. To account for transfection efficiency, an algebraic parameter was generated of the mean fluorescence intensity (MFI) of PEG10-Dendra2 divided by the MFI of IRES-CFP for each event. To generate summary data, the geometric mean of the custom algebraic parameter was generated for each sample.

### Human protein atlas

Data on RTL8 and PEG10 mRNA expression in human tissues was taken from the RNA Expression Consensus dataset containing data from The Human Protein Atlas and GTEx transcriptomics. RNA consensus tissue gene data was downloaded from the Human Protein Atlas on June 17 2023 and is based on The Human Protein Atlas version 23.0 and Ensembl version 109.

### Statistics

Error bars represent mean ± SEM for all Figs. All statistical analysis was performed using GraphPad Prism software. Standard one-way ANOVA was corrected for multiple comparisons by Dunnett's test as recommended. Standard two-way ANOVA was corrected for multiple comparisons by Šídák's test as recommended. For all Figs, statistical tests are listed in the Fig legend and *p<0.05, **p<0.01, ***p<0.001, and ****p<0.0001.

## Results

### PEG10 is spontaneously released as virus-like particles in some human cell lines

Endogenous and transfected human PEG10 have been observed to form VLPs that are released into cell culture medium [5, 16, 17], but the extent of endogenous PEG10-derived VLP production in human cells was unknown. To generate a more complete understanding, we quantified VLP abundance in conditioned medium from a range of cultured human cell lines. We tested the abundance of PEG10 in cell lysate (Fig 1A and 1B, S1 Fig in S1 File) and conditioned medium following enrichment of extracellular particles by ultracentrifugation (Fig 1C and 1D, S1 Fig in S1 File). PEG10 abundance in lysate and the VLP fraction was measured by western blot using a polyclonal antibody which detects the short ('gag') and long ('gag-pol') forms of PEG10, enabling distinct quantitation of both protein species in each cell line.

All cell lines showed detectable levels of PEG10 gag and gag-pol protein in cell lysate (Fig 1A and 1B). In contrast, gag and gag-pol proteins were only detectable in the VLP fraction of HepG2 and hTR-8 cells (Fig 1C and 1D). HepG2 cells had the highest lysate abundance and VLP production of all cell lines tested. However, across all cell lines, lysate levels of PEG10 did

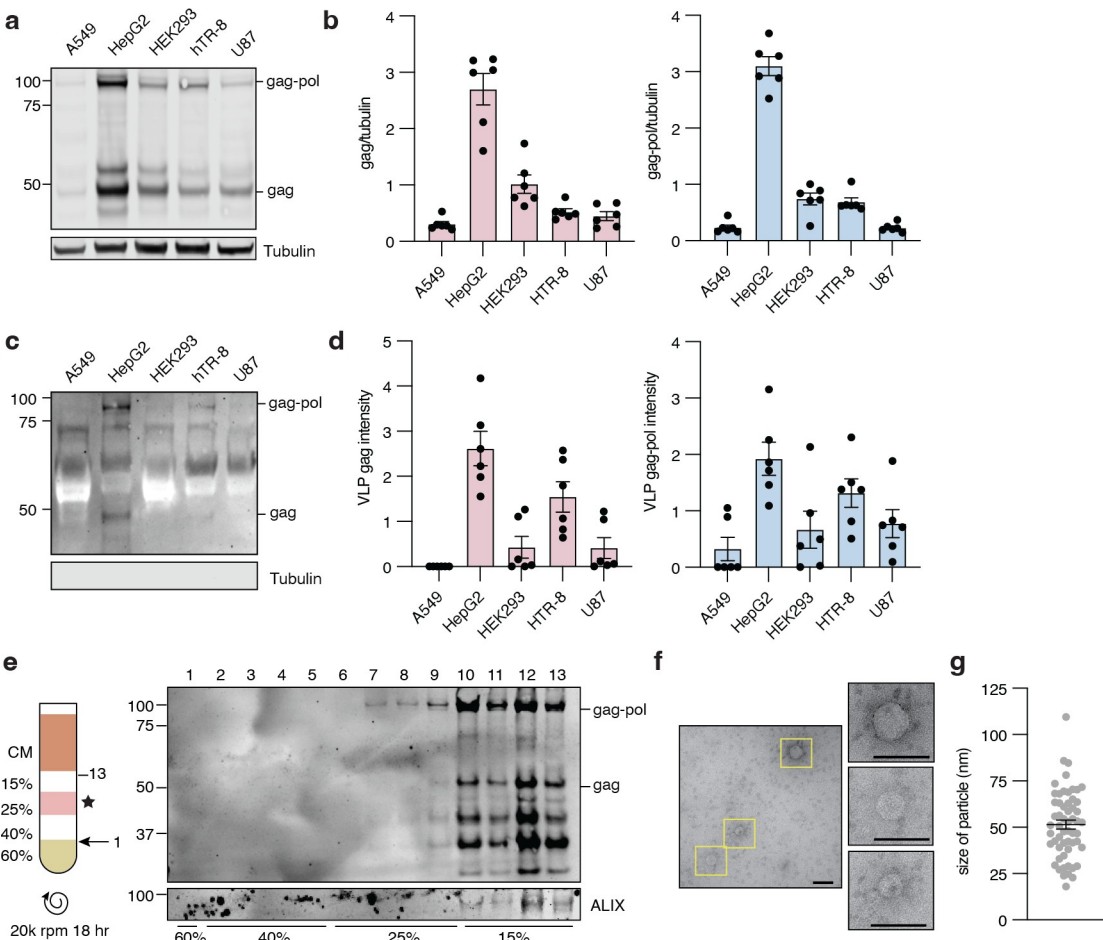

**Fig 1. PEG10-derived VLPs are naturally released by human cells. a)** Representative western blot showing PEG10 abundance in lysate from A549, HepG2, HEK293, hTR-8, and U87 cell lines. n = 6 representative experiments. **b)** Quantification of PEG10 gag (left) and gag-pol (right) abundance in lysate. PEG10 bands were normalized to Tubulin, then normalized to the average of that band across all cell lines for each experiment. **c)** Representative western blot showing PEG10 VLP production across cell lines. n = 6 representative experiments. **d)** Quantification of PEG10 gag (left) and gag-pol (right) abundance in VLPs of human cell lines. PEG10 bands were normalized to the average of that band across all cell lines for each experiment **e)** Iodixanol fractionation of conditioned medium from HepG2 cells. Left: schematic of iodixanol gradient preparation and harvest. CM: conditioned medium. Arrow with 1 indicates where needle was inserted to start fraction collection. Star denotes the percentage of iodixanol which demonstrated the beginning of PEG10 detection. Right: western blot of 13 consecutive fractions from iodixanol gradient showing PEG10 and ALIX. Approximate division of iodixanol fractions is denoted at bottom based on color and flowrate of eluate. **f)** Fraction 8 of an iodixanol prep was used for negative stain electron microscopy of particles. Shown is one representative image of 50 highlighting three VLPs in yellow boxes, magnified at right. Scale bar = 100 nm. **g)** Size measurement of every rounded particle visualized in image set from (f). 57 putative VLPs were measured.

not strongly correlate with VLP abundance (S1 Fig in S1 File). As an alternative hypothesis, HepG2 and hTR-8 cell lines exhibited the highest ratio of gag-pol:gag protein amongst all cell lines tested (S1 Fig in S1 File), indicating a possible role for the gag-pol form of PEG10 in regulation of VLP yield.

Elevated PEG10 protein expression has been implicated in the neurodegenerative disease Amyotrophic Lateral Sclerosis (ALS) and the neurodevelopmental disease Angelman syndrome [17, 28, 38]. We tested for PEG10 VLPs in conditioned media from human iPSC-derived neurons and observed a strong PEG10 gag band (S2 Fig in S1 File). There was a strong cytoplasmic control band present in the ultracentrifuged medium, which is likely derived from

the extracellular matrix necessary for neuron culture, but may represent some level of media contamination with nonspecific cellular fragments.

To further purify PEG10-derived VLPs, a standard iodixanol fractionation was performed. First, fractionation was performed on conditioned medium from HepG2 cells. ALIX, a marker of exosomes [39], was detected in the fractions corresponding to 15% iodixanol (Fig 1E). PEG10 gag-pol and gag were first detected in the fractions corresponding to 25% iodixanol, before ALIX was evident (Fig 1E); however, PEG10 was also abundant in ALIX-positive fractions (Fig 1E). Iodixanol fractionation of conditioned medium from transfected HEK293 cells, which do not normally release PEG10-derived VLPs (Fig 1C), also displayed the presence of HA-tagged PEG10 in the 25% and 15% fractions (S2 Fig in S1 File), indicating that transfected, HA-tagged PEG10 can form VLPs that sediment similarly to endogenously-formed PEG10--derived VLPs.

PEG10-positive fractions from the iodixanol preparation were then prepared for negative stain EM. To minimize the contribution of ALIX-positive exosomes in contaminating EM analysis, the #8 sample from iodixanol fractionation was prepared for imaging. Fraction #8 contained many particles of approximately 50 nm in size (Fig 1F and 1G), consistent with previous EM-based visualizations of PEG10-derived VLPs [5, 16]. Therefore, we conclude that these ~55 nm particles likely represent intact PEG10-derived VLPs.

## RTL8 incorporates into PEG10 VLPs and decreases their abundance in medium

PEG10 is one of a family of closely related gag-like genes in humans that are collectively referred to as the Mart family [29, 40]. Previous studies have implicated a relationship between the *Mart* genes *PEG10* and *RTL8* [5, 38, 41], previously known as *Cxx1* or *FAM127*. *RTL8A*, *RTL8B*, and *RTL8C* are three nearly identical members of the *Mart* family that encode a truncated gag-like protein with high levels of homology to PEG10 [29]. In particular, amino acid alignment of RTL8 with PEG10 shows that RTL8 bears a strong resemblance to the $CA^{NTD}$ portion of PEG10 gag (Fig 2A, S3 Fig in S1 File). At the amino acid level, there is less than 30% identity between RTL8C and PEG10 gag; however, Aphafold modeling revealed a striking similarity in the predicted structures of the two proteins, with RTL8 threading closely to the PEG10 structure in the $CA^{NTD}$ region of gag (S3 Fig in S1 File). Structural comparison of *Homo sapiens* RTL8C and PEG10 gag using the DALI protein structure comparison server showed a high level of probable homology with a z score of 13.9 despite the small size of RTL8 [34].

Due to the structural similarity [29] and reported interactions between RTL8 and PEG10 [42, 43], we hypothesized that RTL8 may incorporate into PEG10 VLPs. First, iodixanol-fractionated HepG2 VLPs were probed with an anti-RTL8 antibody, but no signal was observed at the expected molecular weight (not shown). Next, we tried an overexpression approach. HEK293 cells were used as a model because of their ease of transfection and their lack of natural PEG10 VLP production, which would interfere with interpretation of VLP formation from transfected constructs. FLAG-RTL8c was co-expressed in HEK293 cells with empty vector or with HA-PEG10 gag-pol, followed by analysis of the VLP fraction in ultracentrifuged media by western blot. FLAG-RTL8c was only found in the crude VLP fraction when HA-PEG10 was co-expressed (Fig 2B), indicating that RTL8 is unable to independently form VLPs but can incorporate into VLPs produced by PEG10.

To determine an approximate ratio of PEG10:RTL8 abundance in ultracentrifuged VLPs, HA-tagged PEG10 was expressed along with an HA-tagged RTL8c. The relative abundance of HA-RTL8c was decreased in the VLP fraction compared to cell lysate: upon co-expression,

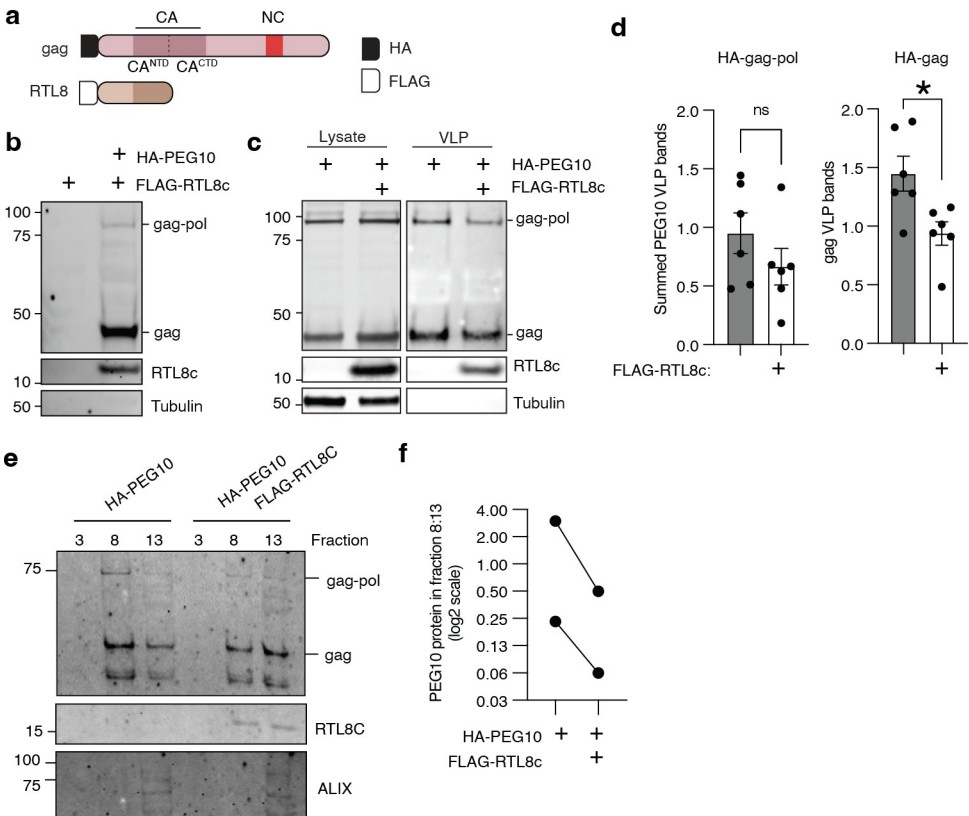

**Fig 2. RTL8 incorporates into PEG10 VLPs and decreases the efficiency of release. a)** Schematic of PEG10 with RTL8. **b)** Representative western blot measuring FLAG-RTL8c presence in the VLP fraction. FLAG-RTL8c was co-expressed without or with HA-PEG10. FLAG-RTL8c release in the VLP fraction is dependent upon co-expression of HA-PEG10. **c)** Representative western blot showing HA-PEG10 lysate (left) and VLP (right) signal without and with FLAG-RTL8c co-expression. **d)** Quantification of HA-PEG10 VLP signal with FLAG-RTL8c co-expression. Left: For PEG10 gag-pol overexpression, gag and gag-pol signal was summed to calculate total PEG10 VLP signal. Right: For PEG10 gag overexpression, gag signal was quantified. For both, PEG10 VLP bands were first normalized to lysate levels of PEG10 as a transfection control, then to the average of all samples for each experiment. Significance was determined by T-test. **e)** Western blot from iodixanol preparation of PEG10-transfected or PEG10 and RTL8c co-transfected HEK293 cells. Cells were transfected, media was changed, and 24–48 hours later, conditioned medium was harvested for iodixanol fractionation and western blot of HA-PEG10, FLAG-RTL8, and ALIX. One quarter of sample #13 was run in comparison to sample #8 to avoid overloading gel. n = 2 independent experiments. **f)** Quantitation of the results from (e) comparing the abundance of PEG10 in fraction 8 and fraction 13. All PEG10 bands were summed and a ratio of abundance was generated for each sample. n = 2.

RTL8 is found at approximately 15% the abundance of PEG10 in the lysate, but is approximately 5% the abundance of HA-PEG10 in the VLP fraction (S3 Fig in S1 File). Together, they indicate that either RTL8 incorporation into PEG10-derived VLPs is inefficient, or that only low levels of RTL8 incorporation are tolerated before VLP formation or release is impossible.

To test whether RTL8 co-transfection specifically altered extracellular abundance of PEG10, both compartments were tested by western blot. Co-transfection of HA-tagged PEG10 with FLAG-tagged RTL8c caused no gross changes to gag-pol or gag abundance in cell lysate (Fig 2C, left). In contrast, co-expression of RTL8c decreased the overall abundance of PEG10 VLPs in conditioned media (Fig 2C, right). This apparent difference was not significant when both gag and gag-pol bands were summed to approximate VLP abundance (Fig 2D, left), but when just the gag form of PEG10 was transfected, the addition of RTL8c significantly decreased VLP abundance in the medium (Fig 2D, right).

Next, relative PEG10 abundance was compared by iodixanol preparation of conditioned medium from HEK cells transfected with HA-PEG10 or co-transfected with HA-PEG10 and FLAG-RTL8c. There was no PEG10 visible in fraction 3, consistent with HepG2 fractionation (Fig 1E). In contrast, PEG10 gag and gag-pol bands were present in fraction 8 and in fraction 13, which was also positive for the exosome marker ALIX (Fig 2E). Co-transfection with FLAG-RTL8c resulted in a visible RTL8 band in both fractions 8 and 13 (Fig 2E), confirming that it co-sediments with PEG10-positive VLPs. Cells co-transfected with RTL8 had comparatively less PEG10 in fraction 8, which was especially evident when directly comparing PEG10 abundance in fraction 8:13 for each transfection condition (Fig 2F). Therefore, co-transfection with RTL8 has a specific effect on the abundance of PEG10 in fraction 8 of iodixanol preparation.

Negative stain EM was performed on iodixanol fractions of HEK293 cells transfected with PEG10 and RTL8. Like in HepG2 cell preparations, fraction #8 was prepared for negative stain as it exhibited PEG10 by western blot without detectable contamination by the exosome marker ALIX (S2 Fig in S1 File). In the absence of transfection, isolated fractions from HEK293 cells had particles of approximately 25 nm in size (Fig 3A and 3B), in contrast to HepG2 cells (Fig 1F and 1G). Upon transfection with HA-PEG10, there appeared many particles of 40–150 nm in size (Fig 3A and 3B). Strikingly, these particles of larger size almost entirely disappeared from fraction #8 when cells were co-transfected with HA-PEG10 and FLAG-RTL8c (Fig 3A and 3B). When size distribution of particles was plotted for all EM samples, including HepG2 cells, only HepG2 cells and HEK293 cells transfected with HA-PEG10 had a large proportion of their VLPs that were larger in size (Fig 3C). Using untransfected HEK293 cells as a guide, a population gate was made at >35 nm of particle size. When this gate was applied to the size distribution of all measured particles, HA-PEG10 transfected cells and HepG2 cells had over 50% of their VLPs larger than 35 nm in size, while untransfected HEK293 cells, and FLAG-RTL8c co-transfected cells had less than 10% of their particles larger than 35 nm (Fig 3D). When only these large particles were considered, the average size of particles from HA-PEG10 transfected HEK293 cells and untransfected HepG2 cells was remarkably consistent at approximately 57 nm in diameter (Fig 3E).

Fraction #13, which showed PEG10 presence by western blot but was contaminated by ALIX (Fig 2E), was also prepared for EM staining. In contrast to fraction #8, these particles were almost exclusively smaller in size (S5 Fig in S1 File) and showed no distinction in particle size or number between any HEK293 cell sample (S5 Fig in S1 File). It is possible that the #13 fraction is too contaminated with exosomes or other particles to identify unique PEG10-derived VLPs. Conversely, it is also possible that this fraction consists of exosomes or other particles that contain PEG10 protein that are indistinguishable with particles devoid of PEG10 protein.

## RTL8 sequesters PEG10 in cells

The decrease in VLP abundance of PEG10 upon co-transfection suggested that RTL8 may inhibit VLP formation, release, or stability. Further quantification of PEG10 abundance in cell lysate from Fig 2C showed a significant increase in intracellular PEG10 when PEG10 gag-pol was co-expressed with RTL8c (S4 Fig in S1 File). To examine the retention of intracellular PEG10 using an orthogonal approach, a flow cytometry-based fluorescent reporter of PEG10 abundance was used. PEG10 was fused at the C-terminus to the fluorophore Dendra2 [44] as a reporter for PEG10 abundance, with an IRES-eCFP cassette as a transfection efficiency control (Fig 4A). Fusion of PEG10 with Dendra2 did not interfere with VLP production, as a VLP prep of PEG10-Dendra2 was positive for PEG10 (S4 Fig in S1 File). Co-expression of the PEG10 reporter construct with FLAG-RTL8c is associated with increased intracellular fluorescence as measured by flow cytometry (Fig 4B), consistent with intracellular accumulation of

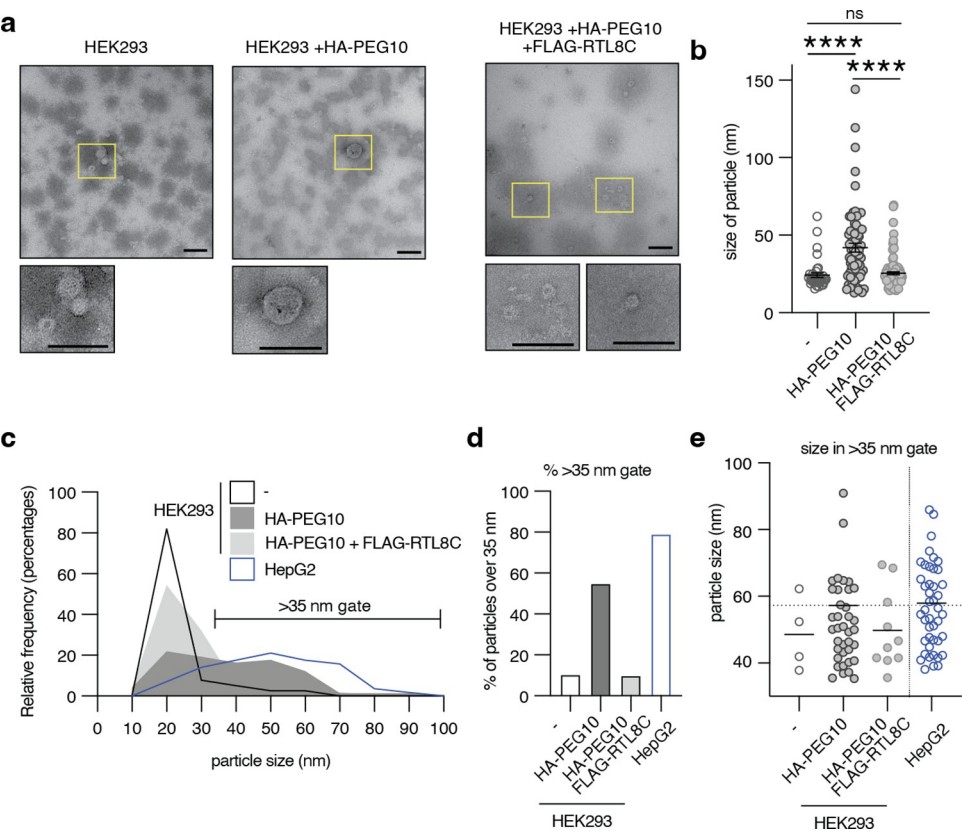

**Fig 3. PEG10 particles disappear upon co-transfection with RTL8c. a)** Representative images of particles from HEK293 cells with and without transfection. VLPs were isolated by iodixanol fractionation and the 25% fraction (#8) was used for negative stain. At least 20 images were collected of each condition. Scale bars are 100 nm. Yellow boxed VLPs are shown at the right at higher magnification. **b)** Particle diameter ('size') was measured for as many VLPs as possible for each set of images. n = 39 untransfected HEK; 73 PEG10-transfected HEK, 103 PEG10- and RTL8-cotransfected HEK VLPs measured. Statistics were determined via unpaired, two-tailed T tests. **c)** Histogram showing relative frequency of VLPs at different sizes from each sample. Untransfected HEK cells are outlined in black, PEG10-transfected cells are in dark grey, PEG10- and RTL8C-cotransfected cells are in light grey, and HepG2 cells are outlined in blue. Data is cut off at 100 nm. **d)** Percentage of VLPs that are over 35 nm in size for each sample. **e)** Size in diameter (nm) of VLPs that are over 35 nm for each sample. Dotted line on the y-axis is at 57.3 nm which was the average for PEG10-transfected HEK cells and is nearly identical to HepG2 average (57.9 nm). For (a-e), n = 1 EM preparation of a representative iodixanol fractionation.

PEG10 and supporting the hypothesis that RTL8 inhibits either PEG10 VLP assembly or release.

Given that RTL8 closely resembles the N-terminus of PEG10, we speculated that RTL8 restricts VLP abundance by mimicking PEG10's N-terminus and competitively incorporating into a growing capsid. In this model of interference, co-transfection of a truncated PEG10 capsid$^{NTD}$ with full-length PEG10 should similarly restrict VLP abundance. PEG10-Dendra2 was co-expressed with an unlabeled capsid$^{NTD}$, resulting in similarly elevated levels of intracellular PEG10-Dendra2 (Fig 4B). These data support a model where RTL8 acts as a competitive inhibitor of PEG10 capsid assembly.

## RTL8 interacts with the N-terminal lobe of the PEG10 capsid domain

To determine whether RTL8 was inhibiting PEG10-derived VLP release from cells through protein-protein interactions, PEG10 was immunoprecipitated from HepG2 cells and blotted

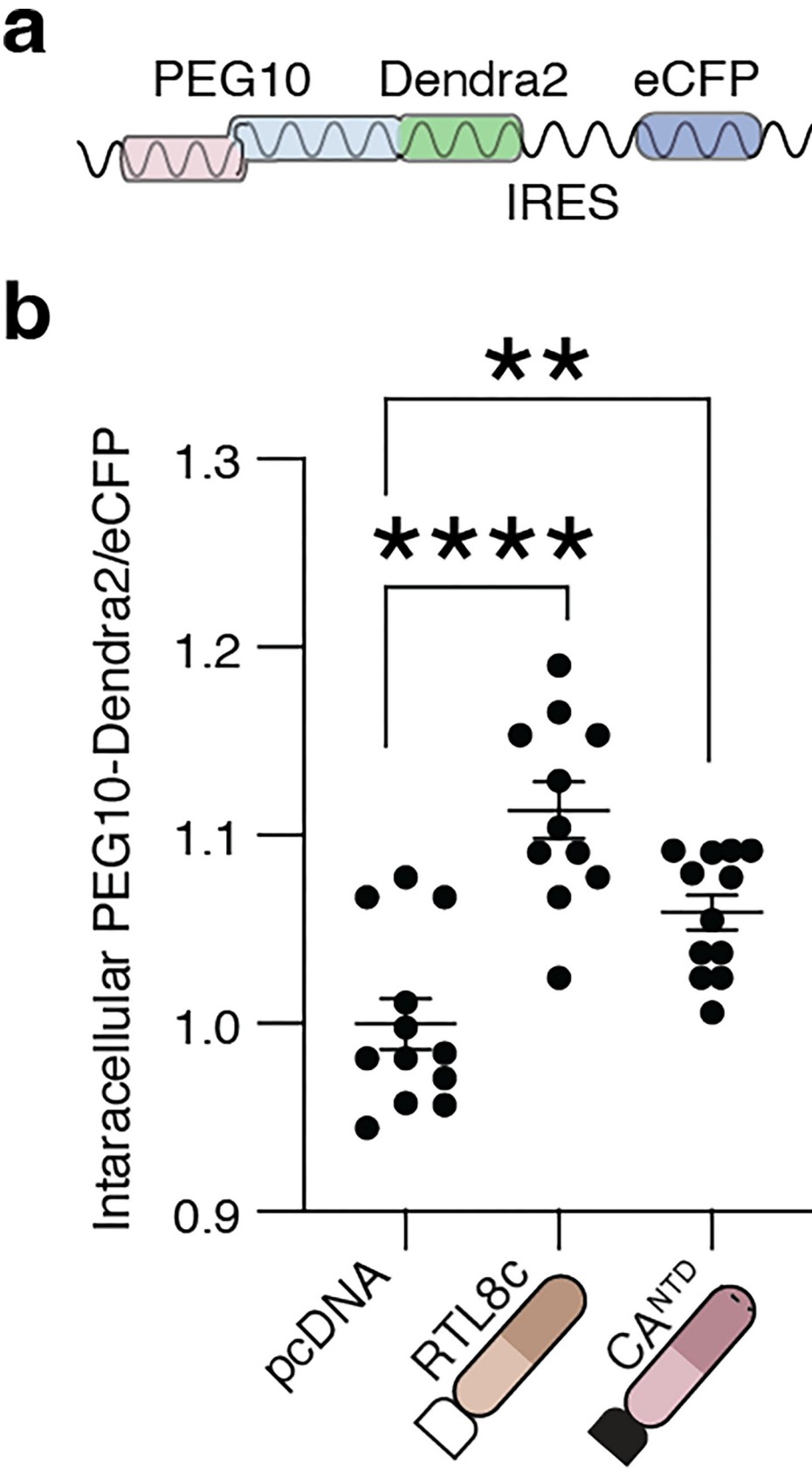

**Fig 4. RTL8c co-transfection increases intracellular retention of PEG10. a)** Design schematic of a flow assay assessing intracellular PEG10 signal. PEG10 is expressed as a fusion protein with the fluorescent protein Dendra2 at its C-terminus, followed by an IRES-CFP for transfection efficiency control. **b)** Flow cytometry measurement of intracellular PEG10-Dendra2 abundance. PEG10-Dendra2 was co-expressed with either a control vector (pcDNA3.1), FLAG-RTL8c, or HA-capsid$^{NTD}$, and PEG10-Dendra2 was measured to determine its intracellular abundance. Data were analyzed by ordinary one-way ANOVA.

against RTL8. RTL8 was readily co-immunoprecipitated when PEG10 was immunoprecipitated, but not with IgG control (Fig 5A, left), indicating interaction between the two proteins. Overexpressed eGFP was not co-immunoprecipitated with PEG10 (Fig 5A, right), highlighting the specific nature of the interaction.

To further probe the interaction between PEG10 and RTL8, HA-PEG10 and FLAG-RTL8c were expressed in HEK293 cells and crosslinking co-immunoprecipitation was performed.

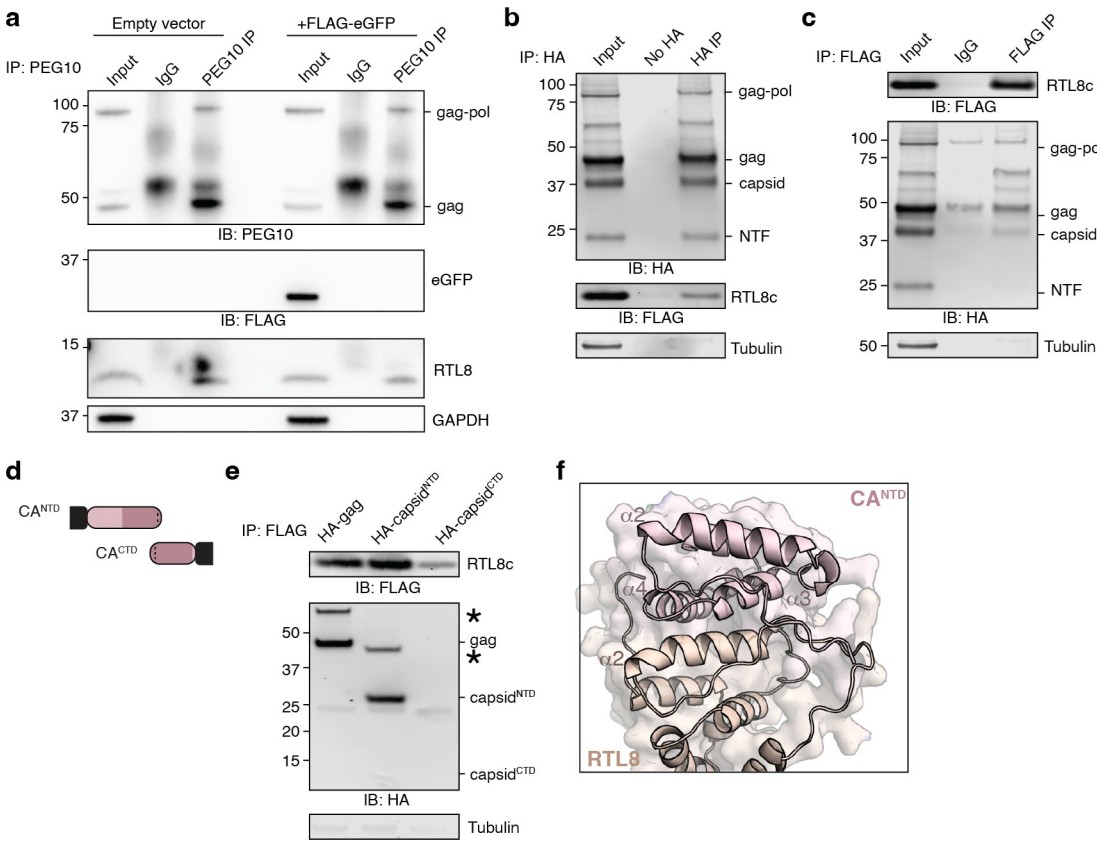

**Fig 5. RTL8 interacts with the NTD lobe of PEG10 capsid. a)** Co-immunoprecipitation of endogenous PEG10 and RTL8 protein from HepG2 cell lysate using a polyclonal PEG10 antibody or IgG control. On the left, cells were transfected with empty vector; on the right, cells were transfected with FLAG-tagged eGFP as a control for nonspecific interactions. **b)** Crosslinking co-immunoprecipitation of RTL8c with PEG10. HA-tagged PEG10 and FLAG-tagged RTL8c were co-expressed, proteins crosslinked, cells lysed, and complexes immunoprecipitated using an HA antibody. No-HA control cells were transfected only with FLAG-RTL8c before HA-immunoprecipitation. **c)** Crosslinking co-immunoprecipitation of PEG10 with RTL8c. FLAG-tagged RTL8c and HA-tagged PEG10 were co-expressed and prepared as in Fig 2B. Complexes were immunoprecipitated using a FLAG antibody or an IgG isotype control antibody. **d)** Diagram of PEG10 constructs including HA-capsid$^{NTD}$ and HA-capsid$^{CTD}$ truncations tested for interaction with FLAG-RTL8c. N-terminal HA tag is shown in black. **e)** Representative immunoprecipitations of FLAG-RTL8c showing co-immunoprecipitation of HA-gag and HA-capsid$^{NTD}$. Asterisk indicates an additional high molecular weight product positive for both HA and FLAG. Due to low relative expression of HA-capsid$^{CTD}$, an additional experimental replicate was performed with an overabundance of HA-capsid$^{CTD}$ lysate in S5 Fig in S1 File. **f)** Alphafold modeling of RTL8 with PEG10 shows contacts between RTL8 and the CA$^{NTD}$ of PEG10. PEG10 is shown in pink, and RTL8 in yellow. Alpha helices of each protein are numbered.

HA-PEG10 was visible as four distinct bands representing gag-pol, gag, and two self-cleavage products generated via activity of the PEG10 protease domain [17, 45, 46] (Fig 5B). The 37 kDa band represents a retrovirus-like capsid fragment, and the 22 kDa band reflects a protein fragment consisting of the N-terminal ~100 amino acids of the protein (referred to as the 'N-terminal fragment', or 'NTF') [17]. Upon immunoprecipitation of HA-PEG10, we observed co-immunoprecipitation of FLAG-RTL8c (Fig 5B). When we performed the reciprocal experiment (IP-FLAG), we observed co-immunoprecipitation of HA-PEG10 gag-pol, gag, and capsid, but not the shorter proteolytically cleaved N-terminal fragment (Fig 5C), indicating that while gag and capsid are sufficient to interact with RTL8c, the very N-terminal region of PEG10 is insufficient.

Retroelement capsid proteins have two distinct lobes: the N-terminal lobe (capsid$^{NTD}$) and C-terminal lobe (capsid$^{CTD}$) [47–49]. During retroelement capsid assembly, NTD lobes from capsid monomers assemble into penta- and hexameric cones, which are held together by homotypic CTD:CTD interactions on adjacent cones [50–54]. Based on sequence alignment and structural modeling, PEG10 capsid closely resembles that of the ancestral Ty3 retrotransposon, including the defined NTD and CTD lobes (S6 Fig in S1 File). Indeed, recent structural studies confirm that PEG10 capsid$^{CTD}$ closely resembles ancestral Ty3 capsid and is capable of dimerization [55].

PEG10 capsid truncation constructs representing the N- and C-terminal lobes (HA-capsid$^{NTD}$ and HA-capsid$^{CTD}$) were generated to map out the location of the PEG10:RTL8 interaction (Fig 5D). Constructs were co-expressed with FLAG-RTL8c, and FLAG was immunoprecipitated. FLAG-RTL8c interacts with HA-PEG10 gag and capsid$^{NTD}$, but not capsid$^{CTD}$ (Fig 5E, S6 Fig in S1 File), indicating that the NTD lobe is both necessary and sufficient for interaction with RTL8. Co-immunoprecipitation also resulted in the appearance of a higher molecular weight band that stained positive for both FLAG and HA (Fig 4E, S6 Fig in S1 File, asterisks), which is likely to be the crosslinked dimer of PEG10 with RTL8.

It was surprising that FLAG-RTL8c was unable to co-immunoprecipitate the small N-terminal fragment (NTF) of PEG10 (Fig 5C) given that the slightly larger HA-capsid$^{NTD}$ construct was both necessary and sufficient (Fig 5E). Truncation constructs of capsid were generated that mimic the natural self-cleavage of PEG10 and were co-expressed with FLAG-RTL8c (S7 Fig in S1 File), followed by co-IP. Neither HA-NTF nor HA-CTF were capable of interaction with FLAG-RTL8c (S7 Fig in S1 File), consistent with results in Fig 5E and suggesting that the entirety of the capsid$^{NTD}$ lobe is necessary for interaction between PEG10 and RTL8.

Alphafold multimer modeling of a putative PEG10:RTL8 interaction further supports the role of the PEG10 NTD in facilitating an interaction of the two proteins. Consistent with an NTD:NTD-like model of binding, an Alphafold multimer prediction showed close interaction between RTL8 and the NTD of PEG10 (Fig 5F). Residues predicted to interact were predominantly in the second α-helix of RTL8, located between residues 41–55, and the third and fourth α-helices of PEG10 gag, located between residues 110–135 (Fig 5F). In contrast, PEG10 self-association appears to favor a CTD:CTD-like interaction, with NTDs left unbound (S7 Fig in S1 File). If PEG10 CTDs are removed to force an NTD:NTD-like interaction, the modeled dimer closely resembles the RTL8:PEG10 interface, with the third and fourth α-helices of both NTD domains closely interacting (S7 Fig in S1 File). An overlay of this NTD:NTD model on top of the RTL8:PEG10 model shows agreement of the two structures, with the third α-helix of PEG10 resting within the second α-helix of RTL8 (S7 Fig in S1 File). Based on these data, as well as the structural homology of PEG10 capsid$^{NTD}$ and RTL8, we conclude that the PEG10:RTL8 interaction likely resembles the homotypic capsid NTD:NTD interactions typical of the retroelement capsid assembly process [51].

## The intracellular retention of PEG10 by RTL8 shows species, but not paralog, specificity

Three paralogs of *RTL8* (*RTL8a*, *b*, and *c*) are all found on the X-chromosome and share >92% identity [29]. We compared all three *RTL8* gene products by flow cytometry and saw no differences in their association with intracellular PEG10 abundance (Fig 6A), indicating that each form of RTL8 is equally associated with elevated intracellular PEG10.

One hallmark of gag-based restriction of retroelement capsids is the specificity of interaction. In mice, alleles of the gag-like gene *Fv1* show differential ability to inhibit unique strains of MuLV [56–58]. Like humans, mice express three *Rtl8* paralogs which share considerable homology to human *RTL8* genes as well as *Peg10*, which diverges from human in the expansion of a proline and glutamine-rich region near the C-terminus [29]. Mouse FLAG-Rtl8b, which shares the highest homology to human RTL8c (Fig 6B), bound less efficiently to human PEG10 by co-immunoprecipitation (Fig 6C). Further, co-expression of mouse Rtl8b had no effect on the intracellular retention of human PEG10-Dendra2 by flow cytometry (Fig 6D). Conversely, mouse PEG10 was capable of binding both human and mouse FLAG-RTL8 (Fig 6E) but its intracellular levels were not increased (Fig 6F). In conclusion, the RTL8 effects on PEG10 intracellular and VLP abundance are only evident using human genes.

## High levels of RTL8 expression are observed in a low-VLP yield cell line

Given the ability of RTL8 to inhibit PEG10-derived VLP release upon transfection, we speculated that endogenous expression of RTL8 may correlate with the ability of cells to spontaneously release VLPs. Endogenous protein levels of RTL8 in the cell lines tested in Fig 1 were very low and could only be visualized and quantified using sophisticated western blot imaging equipment (Fig 7A and 7B). We found that two of the cell lines with the lowest RTL8 expression had high VLP yields, while cells with the highest level of RTL8 protein had no detectable PEG10-derived VLPs in conditioned medium (Fig 7C).

To explore a potential correlation further, a directed approach to cell line selection was taken. Tissue expression profiles of *RTL8C* and *PEG10* mRNA from the Human Protein Atlas [59] were examined and the top ten enriched tissues for each gene were plotted. Both *RTL8* and *PEG10* transcripts were particularly enriched in brain tissues (Fig 7D, yellow bars). Western blot of postmortem human brain samples also showed detectable levels of PEG10 and variable RTL8 protein expression in the cortex (Ctx) and less in the substantia nigra (SN) (Fig 7E). Based on these findings, a second panel of cell lines derived from brain tissue was assembled for a secondary analysis of VLP production and RTL8 abundance. A variety of cell types were chosen, including astrocyte-like cells (CCF-STTG1), glial and glial-like fibroblastic cells (M059K and T98G), and neuroblast cells (BE(2)-M17 and SK-N-SH). PEG10 levels in lysate were low, but detectable, in all brain cell lines compared to HepG2 cells (Fig 7F). In contrast, all cell lines had higher RTL8 than HepG2 cells (Fig 7F). When VLP fractions were prepared, HepG2 cells displayed high levels of PEG10, whereas brain cell lines had faint or no discernable PEG10. M059K cells exhibited only a faint gag band, and BE(2)-M17 cells showed faint, but readily discernable, gag and gag-pol (Fig 7G). Therefore, while these data suggest that high levels of PEG10 and low levels of RTL8 are associated with increased VLP abundance, further work is needed to understand the determinants of PEG10-derived VLP release from cells.

## Natural production of PEG10 VLPs can be enhanced through modulation of RTL8

The utility of PEG10-derived VLPs for biotechnological use relies on an ability to produce VLPs in high quantities. The data presented here support a model whereby RTL8 inhibits

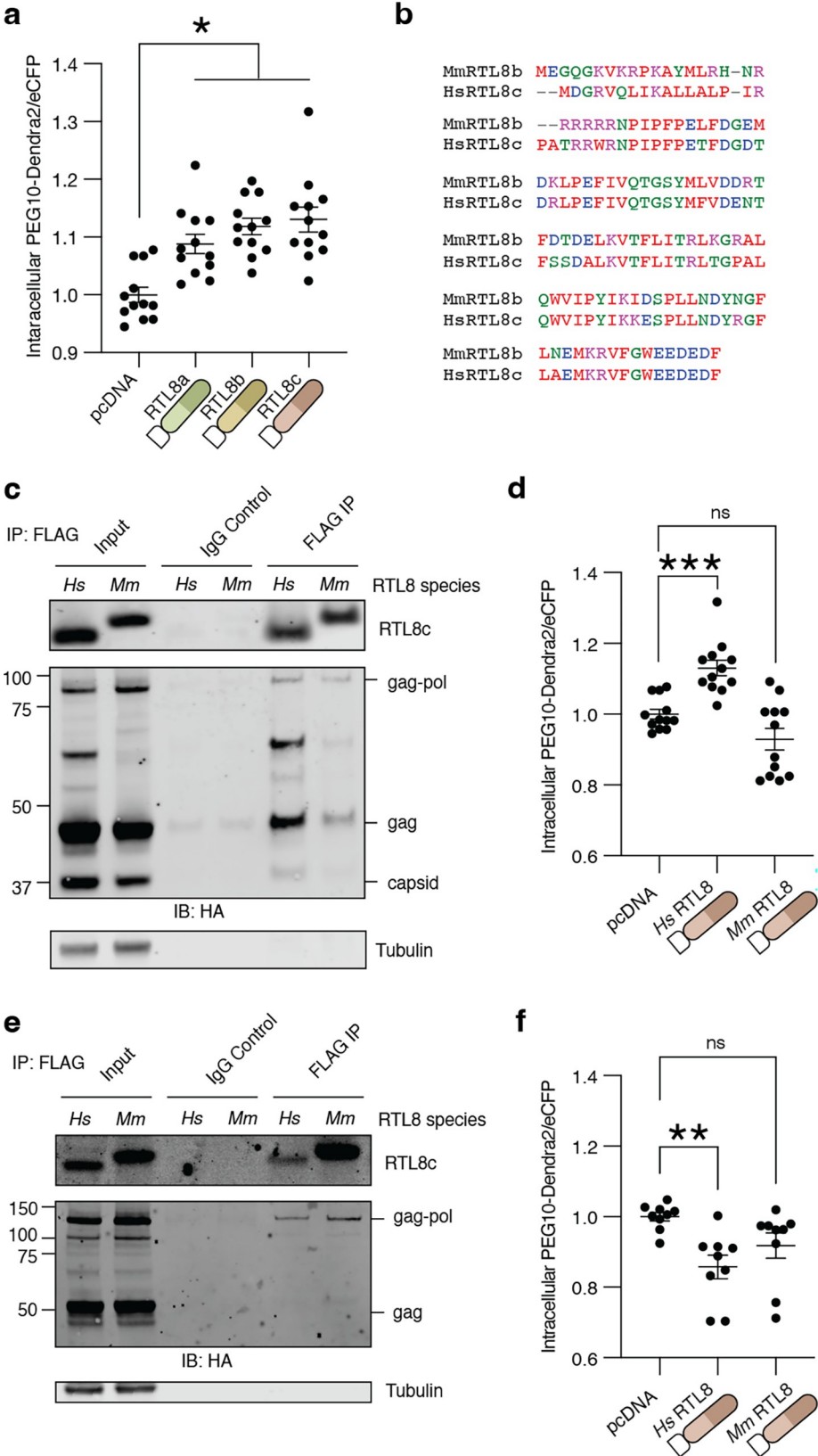

**Fig 6. RTL8 binding and restriction of PEG10 VLP release is species-specific. a)** Normalized PEG10-Dendra2 abundance upon control vector, or FLAG-RTL8a, FLAG-RTL8b, or FLAG-RTL8c co-expression. Data were analyzed by ordinary one-way ANOVA with column means compared to mock transfected. **b)** Alignment of murine and human RTL8 demonstrating high level of conservation. Colors denote amino acid properties. **c)** Co-immunoprecipitation of murine or human FLAG-RTL8 with HA-tagged PEG10. n = 2 independent experiments. **d)** PEG10-Dendra2 was co-expressed with control, FLAG-*Hs*RTL8c, or FLAG-*Mm*Rtl8b and intracellular abundance was measured by flow cytometry. Data were analyzed as in Fig 2F. **e)** Co-immunoprecipitation of murine or human FLAG-RTL8 with murine HA-tagged Peg10. n = 1. **f)** Murine Peg10-Dendra2 was co-expressed with control, FLAG-*Hs*RTL8c, or FLAG-*Mm*Rtl8b and intracellular abundance was measured by flow cytometry. Data were analyzed as in Figs 2F and 4D. Data shown are from 3 independent experiments.

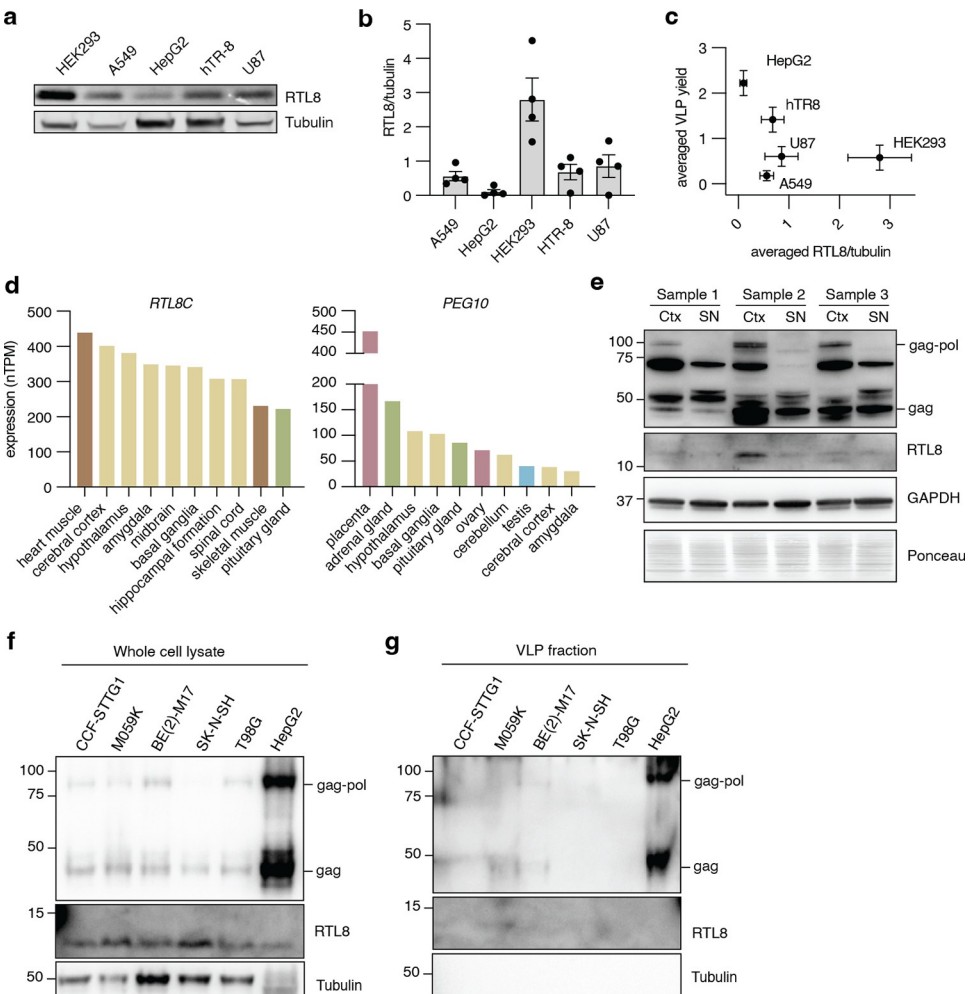

**Fig 7. RTL8 levels are low in cell lines with high VLP abundance, and high in a cell line with low VLP yield. a)** RTL8 levels in cell lysate for cell lines tested in Fig 1. **b)** Quantification of RTL8 abundance in Fig 5A. n = 4 independent experiments. RTL8 was normalized to Tubulin, then to the average of RTL8 across all cell lines for each experiment. Error bars depict SEM. **c)** Intracellular RTL8 levels plotted against VLP abundance. For all cell lines tested, RTL8 was plotted against VLP abundance, with SEM for n = 4 biological replicates. **d)** Human Protein Atlas data of tissue expression profiles for human *RTL8C* and *PEG10*. The Human Protein Atlas consensus gene expression dataset was plotted for the top 10 tissues and color coded for the tissue type. **e)** Western blot of PEG10 and RTL8 protein expression in three postmortem human brain samples. Ctx: Cortex; SN: substantia nigra. **f)** PEG10 and RTL8 levels in cell lysate for a panel of human brain cell lines with HepG2 cells as positive control for VLP-producing cells. **g)** PEG10-derived VLP abundance in the same cell lines.

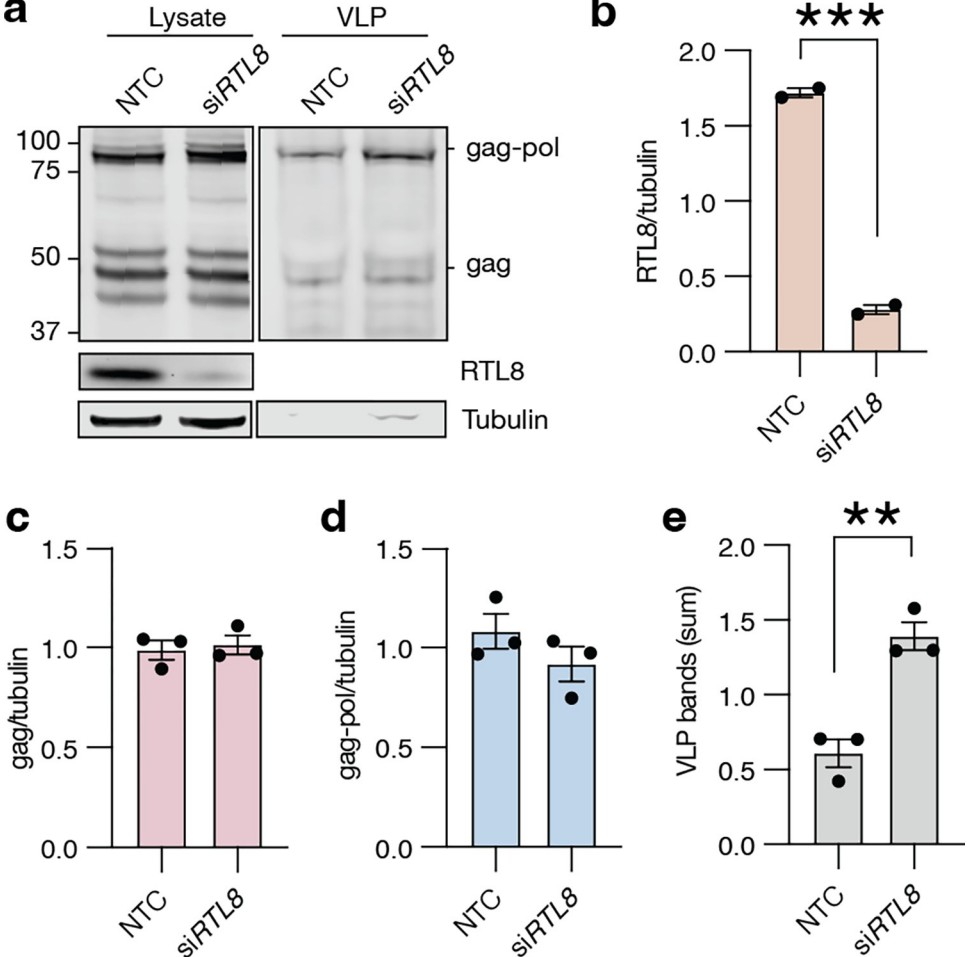

**Fig 8. Modulation of *RTL8* in HepG2 cells is sufficient to alter the production of PEG10-derived VLPs. a)** HepG2 cells were transfected with siRNA against *RTL8* and conditioned medium was harvested 48 hr later for VLP western blot. n = 3 independent experiments. **b)** Quantitation of RTL8 abundance from Fig 6A. RTL8 bands were normalized to Tubulin, then to the average RTL8 across each experiment. **c-d)** Quantification of PEG10 gag (c) or gag-pol (d) abundance in cell lysate. PEG10 was normalized to Tubulin, then to blot average. For b-e, data were analyzed by Student's t-test. **e)** Quantitation of VLP abundance from Fig 6A. PEG10 VLP signal was measured as the sum of gag-pol and gag band intensities normalized as described in Methods.

PEG10-derived VLP release from cells (Fig 2); therefore, modulation of RTL8 may improve VLP yield for the purposes of large-scale production. We sought to increase natural levels of PEG10 VLP production by silencing *RTL8* expression in HepG2 cells, which spontaneously release PEG10 VLPs (Fig 1). Because of the similarity between *RTL8a*, *RTL8b*, and *RTL8c*, we sought to silence all RTL8 genes simultaneously with an siRNA approach. *RTL8* siRNA robustly diminished RTL8 protein levels in cell lysate (Fig 8A and 8B) without changing lysate levels of PEG10 (Fig 8A and 8D). However, *siRTL8*-treated cells showed approximately twice as much PEG10 in the VLP fraction (Fig 8A and 8E), suggesting that the effect of RTL8 is to specifically regulate VLPs. This effect was cell-type specific, as modulation of hTR-8 cells with the same siRNA construct showed no change to VLP abundance (S8 Fig in S1 File). In conclusion, the abundance of PEG10-derived VLPs is remarkably cell-type specific and likely involves many factors, including PEG10 expression level, *RTL8* expression, and other regulatory components that remain poorly understood.

## Discussion

*PEG10* is a unique human gene. Being of retroelement origin, it retains many virus-like features including the ability to form virus-like particles that are released from the cell [5, 16, 17]. We undertook a study to explore the regulation of PEG10-derived VLP formation and release. We found that the ability to spontaneously release PEG10 VLPs was shared amongst multiple human cell lines and was not simply due to expression levels of PEG10 protein. In exploring this regulation further, we found that the closely related gene product RTL8 could be incorporated into PEG10-derived VLPs and decreased their abundance. PEG10 and RTL8 interacted, which was dependent on an intact N-terminal domain of the PEG10 capsid region. In comparison to PEG10, RTL8 levels were low but detectable in cell lines and human brain tissue. Modulation of RTL8 levels in hTR-8 cells measurably increased PEG10-derived VLP yield, indicating that the interplay of these two gag-like genes contributes to VLP regulation.

Based on the data presented here, we propose that *RTL8* is an inhibitor of PEG10-derived VLP abundance. We propose that RTL8 inhibits VLP formation or release through incorporation into VLPs via RTL8:capsid$^{NTD}$ interactions, which is tolerated at low levels, but decreases VLP abundance in conditioned medium. Further, we propose that high levels of RTL8 incorporation prevent the formation or release of VLPs from the cell, resulting in intracellular accumulation of PEG10.

A major question remaining from our study is the mechanism by which PEG10 VLPs are released. Upon iodixanol gradient centrifugation, PEG10 gag and gag-pol bands were observed in ALIX-positive fractions as well as denser, ALIX-negative fractions. PEG10 may be released from cells in a combination of exosomes as well as a denser, virus-like particle. Visualization of samples by negative stain showed major differences in the particles of these fractions. Particles in fraction #8 were unique in PEG10-rich samples, such as HepG2 cells and HEK cells transfected with HA-PEG10, providing compelling evidence that these large particles of about 57 nm represent PEG10-derived VLPs. In contrast, while fraction #13 had even higher levels of PEG10 by western blot, particles could not be distinguished between any sample. It seems likely that these represent particles secreted by cells that have included PEG10 as cargo, due to the inability to distinguish PEG10-expressing from non-expressing samples. Additional work is necessary to explore the putative regulators of PEG10 release in these two fractions, including ESCRT pathway proteins which facilitate exosome budding [60], as well as budding of retroviral virions [61].

It is notable that RTL8 co-transfection profoundly decreased the number of PEG10-derived VLPs in fraction #8 by negative stain while having a more subtle effect on PEG10 abundance by western blot of fraction #8, and especially by western blot of crude VLP preparations. We propose that RTL8 specifically prohibits the formation of these larger VLPs, but may not influence the incorporation of PEG10 into other, exosome-like particles that are more abundant in fraction #13. It is possible that upon RTL8 incorporation into PEG10 capsid oligomers, the absence of an RTL8 capsid$^{CTD}$ domain prohibits the formation of higher-order assemblies generated through both homotypic CTD:CTD and heterotypic NTD:CTD interactions that are necessary for retroelement capsid assembly [50–54, 62] and would be characteristic of PEG10-derived capsids. Alternatively, the lack of an NC domain in RTL8 may destabilize a heterotypic VLP. For example, mutation of the zinc finger of yeast Ty3 gag disrupts particle assembly [63], and the same could occur when RTL8 incorporates into PEG10-derived particles. A third possibility is that RTL8 incorporation disrupts proper localization of the PEG10 capsid or protein:protein interactions necessary for the specific trafficking and release of VLPs. In support of this hypothesis, recent studies have shown that RTL8 influences the localization and function of UBQLN2 [41], and RTL8 could similarly influence PEG10 localization.

Further subcellular investigation of PEG10 capsid assembly and release is necessary to distinguish these possibilities.

Gag-like genes have been co-opted or exapted from retroelements multiple times in eukaryote evolution to restrict the infective capacity of retrotransposons or retroviruses [18–22, 64]. These genes derived from retroelement *gag* genes and act at the stage of capsid formation, release, or uncoating. The sheep gag-derived gene *enJS56A1* restricts infection by the enJSRV family of viruses by disrupting capsid assembly and release [18, 19]. The mouse *Fv1* gene is a gag-like restriction factor that targets incoming murine leukemiavirus (MuLV) particles by coating the endocytosed virus and preventing disassembly of the capsid [22, 64]. In a mechanism that closely mimics the PEG10:RTL8 relationship, the Ty1 retrotransposon in yeast is tightly regulated by the presence of a cryptic start site within the *gag* open reading frame, which results in the production of a truncated capsid$^{CTD}$ fragment. The truncated fragment prevents homotypic CTD:CTD interactions of intact capsid, thereby limiting Ty1 re-integration [20, 21]. In each system, the *gag*-like genes use an affinity for capsid to inhibit the formation, release, or uncoating of the infectious particle. There are many *gag*-like genes in the human genome [8, 12, 13], but no such *gag*-like restriction mechanisms have been described in humans. Based on our data, we posit that *RTL8* can be included in this list of these 'gag decoys' which act at the step of PEG10 virus-like capsid formation or release.

The biological purpose of RTL8 inhibition of PEG10-derived VLPs remains elusive, as does the role of PEG10-derived VLP formation more generally. High levels of PEG10 have been implicated in the neurological diseases Angelman's syndrome [28] and ALS [17, 38], where regulators of PEG10 abundance (*UBE3A* in Angelman's, and *UBQLN2* in ALS) are mutated and unable to control PEG10 levels. RTL8 expression may be a similar means of limiting the pathological role of PEG10 in neural tissue. The ability of *RTL8* to antagonize PEG10-derived VLPs may be additionally relevant when considering the utility of PEG10 in generating biocompatible mRNA delivery systems, such as SEND [16]. Generation of PEG10-derived VLPs for the purposes of nucleic acid delivery in human cells may be limited in cell lines expressing abundant *RTL8*. For the purposes of maximizing PEG10-derived VLP yield, genetic modification of cell lines with the suppression or deletion of *RTL8* may enhance their utility as production lines.

## Supporting information

**S1 File. All supporting information figures.**
(PDF)

**S2 File. Uncropped blots.**
(PDF)

## Acknowledgments

The authors would like to acknowledge the Shared Instruments Pool of the Department of Biochemistry at CU Boulder (RRID: SCR_018986), the Biochemistry Cell Culture Core Facility (RRID: SCR_018988), and the Biochemistry Flow Cytometry Core Facility (RRID: SCR_019309) for their assistance with shared equipment. The authors would also like to acknowledge the Parker laboratory at CU Boulder for their assistance with neuronal conditioned medium. Electron microscopy was done at the University of Colorado, Boulder EM Services Core Facility in the MCDB Department, with the technical assistance of facility staff.

## Author Contributions

**Conceptualization:** Alexandra M. Whiteley.

**Formal analysis:** Will Campodonico, Harihar M. Mohan, Phuoc T. Huynh, Alexandra M. Whiteley.

**Funding acquisition:** Henry L. Paulson, Lisa M. Sharkey, Alexandra M. Whiteley.

**Investigation:** Will Campodonico, Harihar M. Mohan, Phuoc T. Huynh, Holly H. Black, Cristina I. Lau, Lisa M. Sharkey, Alexandra M. Whiteley.

**Methodology:** Will Campodonico, Harihar M. Mohan, Phuoc T. Huynh, Holly H. Black, Cristina I. Lau, Lisa M. Sharkey, Alexandra M. Whiteley.

**Project administration:** Alexandra M. Whiteley.

**Resources:** Henry L. Paulson, Lisa M. Sharkey.

**Supervision:** Henry L. Paulson, Lisa M. Sharkey, Alexandra M. Whiteley.

**Validation:** Will Campodonico, Harihar M. Mohan.

**Visualization:** Will Campodonico, Harihar M. Mohan, Alexandra M. Whiteley.

**Writing – original draft:** Will Campodonico, Alexandra M. Whiteley.

**Writing – review & editing:** Harihar M. Mohan, Phuoc T. Huynh, Henry L. Paulson, Lisa M. Sharkey, Alexandra M. Whiteley.

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
