## [Decision Letter · Decision Letter 0]

28 Jun 2024

PONE-D-24-18574The gag-like gene RTL8 antagonizes PEG10-mediated virus like particlesPLOS ONE

Dear Dr. Whiteley,

Thank you for submitting your manuscript to PLOS ONE. After careful consideration, we feel that it has merit but does not fully meet PLOS ONE’s publication criteria as it currently stands. Therefore, we invite you to submit a revised version of the manuscript that addresses the points raised during the review process.

Both reviewers and I found the article very interesting. I would ask you to please address all the points that the reviewers made. I have selected the decision "Major revision" because I found the following comment from reviewer 1 valid ""To strengthen the results discussed in the manuscript, the authors should provide conclusive evidence on the integrity of the analyzed particles and data on sedimentation coefficients to support the conclusions of the paper."). I would ask you to please show at least conclusive evidence on the integrity of the particles.

We look forward to receiving your revised manuscript.

Kind regards,

Mauricio Comas-Garcia

Academic Editor

PLOS ONE

Journal Requirements:

2. Thank you for stating the following financial disclosure: "H.M.M., L.M.S. and H.L.P. were supported by NIH R35NS122302. H.M.M. was supported by the Rackham Graduate School Rackham Predoctoral Fellowship. W.C. was supported by the University of Colorado Venture Partners. A.M.W. was supported by the University of Colorado Venture Partners and NIH R01NS131660." 

3. Thank you for stating the following in the Competing Interests section: "W.C. and A.M.W. are authors on a patent related to the work presented in this manuscript."

Additional Editor Comments:

Dear Prof. Alexandra Whiteley,

Thanks for submitting your manuscript. Both reviewers and I found the article very interesting. I would ask you to please address all the points that the reviewers made. I have selected the decision "Major revision" because I found the following comment from reviewer 1 valid ""To strengthen the results discussed in the manuscript, the authors should provide conclusive evidence on the integrity of the analyzed particles and data on sedimentation coefficients to support the conclusions of the paper."). I would ask you to please show at least conclusive evidence on the integrity of the particles.

All the best,

Prof. Mauricio Comas-Garcia

Handling Editor

Reviewers' comments:

Reviewer's Responses to Questions

**Comments to the Author**

1. Is the manuscript technically sound, and do the data support the conclusions?

Reviewer #1: Partly

Reviewer #2: Yes

2. Has the statistical analysis been performed appropriately and rigorously? 

Reviewer #1: No

Reviewer #2: Yes

3. Have the authors made all data underlying the findings in their manuscript fully available?

Reviewer #1: Yes

Reviewer #2: Yes

4. Is the manuscript presented in an intelligible fashion and written in standard English?

Reviewer #1: Yes

Reviewer #2: Yes

5. Review Comments to the Author

Reviewer #1: Summary:

The manuscript submitted by Campodonico et al, entitled "The gag-like gene RTL8 antagonizes PEG10-mediated virus like particles" investigates the mechanism by which PEG10-derived retroviral particles are restricted by RTL8 expression. Retrotransposon elements have been implicated in processes related to placentation and neurodegenerative diseases, becoming important aspects of genetic inheritance and gene adaptation. Structurally, RTL8 resembles the NTD of gag proteins, which interfere with the morphogenesis of PEG10 VLPs. Not surprisingly, expression of RTL8, which comprises the NTD domain and not the CTD domain, interferes with the formation of PEG10-derived VLPs. Overall, the manuscript is well written and the results partly support the conclusions raised. However, there is an important issue that the authors need to resolve before recommending publication of this manuscript in PlosONE. Although density gradients are used to fractionate PLVs, there is no control relating the expected sedimentation coefficient of PLVs and the number of fractions. Stronger evidence to support the conclusions raised by the authors would involve the use of analysis of the VLP-containing fraction content by transmission electron microscopy, determining which fractions contain PEG10-derived VLPs and their integrity.

Major: There is a critical point regarding the content and integrity of PEG10-derived VLPs. Although WB analyses show that the PEG10 content in the Iodixanol gradient, there is no control over the sedimentation coefficient or structural integrity of the expected VLPs. To strengthen the results discussed in the manuscript, the authors should provide conclusive evidence on the integrity of the analyzed particles and data on sedimentation coefficients to support the conclusions of the paper.

Minor:

Line 80: define/introduce RTL8 gene. what is the function of RTL8 genes? what is the length of naturally expressed RTL8? is only the NTD domain expressed? is it cell strain or tissue dependent?

Line 161: ul to ml.

Line 364: indicate what percentage of homology/conservation supports the ambiguous statement "strong resemblance".

Line 429: when the authors say "VLP release", readers might be confused and assume that they mean VLPs released into the medium, similar to virus release without lytic phase. Please revise the wording at this point.

Line 466: This statement could be somewhat confusing, as the term "self-association" could lead the reader to think of the assembly pathway rather than the final protein network structure, so I recommend rewriting this sentence.

Line 476: In comparing the results in Figures 3 and S5, this reviewer assumes that the IP experiments in Figure 3 are performed with conditioned medium and those in Figure S5 with lysate. Is this the case? If so, why are no tubulin bands visible in the HA-Gag and HA-NTD lanes?

Lane 538 and Figure 5c. This reviewer is aware that these experiments are somewhat noisy, so conclusions should be drawn with caution. For example, for this reviewer, Figure 5c does not corroborate that intracellular RTL8 content inversely correlates with PEG10-derived VLPs. What is the correlation coefficient of the fit shown in Figure

5c, what are the errors in these data, and can the authors show these data with corresponding error bars?

Line 543: This reviewer finds it confusing to normalize RTL8 levels across all cell lines. Similar to tubulin, can the authors use another band to normalize the data within each cell line? For example, would it be possible to use another vector expressing an excreted protein as a reference? This might be a better approach, as critical differences could be found depending on the cell line used. What would the ratios be if the data were normalized within each cell line? Could it be that RTL8 levels do not correlate with VLP production?

Reviewer #2: This is a sound study of the human retro element Gag protein expression and VLP assembly, that can be suppressed by the expression of the additional retro element coding for the N terminal part of that Gag only. The authors show convincingly that both gag and gag pol are expressed in the cell and are present in the lysates of different type cells in comparable quantities. However, the amount of the released VLPs appeared to vary strongly with the cell type. The authors have shown that the VLP release is modulated strongly by the level of NTD of Gag expression that co-assembles with Gag and GagPol, but gets incorporated into VLP in small quantities only, and supresses the VLP assembly over all. This is an important finding both for beginning to understand the mechanism of Gag assembly, as well as for the VLP cargo delivery applications.

I wonder if the RTL8 supresses Gag assembly because it lacks the RNA-binding NC domain? The retroviral cationic NC domains in the CTDs of Gag are typically supporting the VLP assembly via RNA binding. Lacking NC and RNA binding the RTL8-incorporated VLPs may be simply very unstable. Therefore, it is worth it to follow up on that study and to maybe test the retro element derived protein containing the C-terminal portion of Gag with its NC domain in its effect on VLP formation. It is possible that this protein will be much better incorporated into VLPs and not as supressing for its assembly.

The paper would improve if the authors could put forward a few hypothesis of the mechanism of supression of the VLP assembly by RTL8 in the discussion.

6. PLOS authors have the option to publish the peer review history of their article (what does this mean?). If published, this will include your full peer review and any attached files.

Reviewer #1: No

Reviewer #2: **Yes: **Ioulia Rouzina

---

## [Author Response · Author response to Decision Letter 0]

22 Aug 2024

Reviewers' comments:

Reviewer's Responses to Questions

Comments to the Author

1. Is the manuscript technically sound, and do the data support the conclusions?

Reviewer #1: Partly

Reviewer #2: Yes

2. Has the statistical analysis been performed appropriately and rigorously?

Reviewer #1: No

Reviewer #2: Yes

3. Have the authors made all data underlying the findings in their manuscript fully available?

Reviewer #1: Yes

Reviewer #2: Yes

4. Is the manuscript presented in an intelligible fashion and written in standard English?

Reviewer #1: Yes

Reviewer #2: Yes

5. Review Comments to the Author

Reviewer #1: Summary:

The manuscript submitted by Campodonico et al, entitled "The gag-like gene RTL8 antagonizes PEG10-mediated virus like particles" investigates the mechanism by which PEG10-derived retroviral particles are restricted by RTL8 expression. Retrotransposon elements have been implicated in processes related to placentation and neurodegenerative diseases, becoming important aspects of genetic inheritance and gene adaptation. Structurally, RTL8 resembles the NTD of gag proteins, which interfere with the morphogenesis of PEG10 VLPs. Not surprisingly, expression of RTL8, which comprises the NTD domain and not the CTD domain, interferes with the formation of PEG10-derived VLPs. Overall, the manuscript is well written and the results partly support the conclusions raised. However, there is an important issue that the authors need to resolve before recommending publication of this manuscript in PlosONE. Although density gradients are used to fractionate PLVs, there is no control relating the expected sedimentation coefficient of PLVs and the number of fractions. Stronger evidence to support the conclusions raised by the authors would involve the use of analysis of the VLP-containing fraction content by transmission electron microscopy, determining which fractions contain PEG10-derived VLPs and their integrity.

Major: There is a critical point regarding the content and integrity of PEG10-derived VLPs. Although WB analyses show that the PEG10 content in the Iodixanol gradient, there is no control over the sedimentation coefficient or structural integrity of the expected VLPs. To strengthen the results discussed in the manuscript, the authors should provide conclusive evidence on the integrity of the analyzed particles and data on sedimentation coefficients to support the conclusions of the paper.

We thank the reviewer for this comment and for the push to visualize these particles. Below, we summarize our work to analyze VLP-containing fractions by negative stain electron microscopy, which are included in Figures 1f-g, Figure 3, and Figure S5. The text discussing the results of these experiments are on lines 375-379, and 453-537. Together, we think that these data improve our manuscript and the assertion that we are observing changes to VLP dynamics upon RTL8 co-expression. 

Initially, we had tried negative stain EM on crude VLP preparations and found it impossible to distinguish PEG10-derived VLPs from exosomes present in conditioned and even unconditioned media.

 We tried negative stain imaging again on iodixanol-prepared fractions and had more success. We isolated conditioned medium from HepG2 cells, which naturally produce VLPs, or HEK293 cells, which do not. We also isolated conditioned medium from HA-PEG10 transfected cells and HA-PEG10/FLAG-RTL8C co-transfected cells. We took fraction #8 and #13 from our iodixanol preparations, which corresponded to the 25% fraction (purer, but less PEG10 signal by western blot) and the 15% fraction (which had much more PEG10, but was also positive for ALIX.

We found that it was impossible to distinguish PEG10-derived VLPs in the 15% fraction (Figure S5), as was the case previously with the crude VLP preparation from sucrose cushion. However, in the 25% iodixanol fraction, we found that there were abundant ~55 nm sized particles only in samples expected to release PEG10 VLPs (Figure 1f-g, Figure 3). Furthermore, we found that these particles were effectively eliminated by co-expression with RTL8C (Figure 3).

These data are in agreement with previous literature showing negative stain EM of PEG10-derived particles. Our measurement of ~55 nm particles is in line with Abed et al. (60-80 nm) and Segel et al. (50-80 nm). These particles are smaller than those reported in Pandya et al. (100-150 nm), which could be due to sample source. We isolated particles from HEK cells and HepG2 cells, while Pandya et al. isolated vesicles from iPSC-derived neurons. Our experiments expand on these studies with quantitation and controls. A limitation of the previous literature was that there were limited negative controls for contaminating exosomes or other particles released in media (Pandya and Abed), making it difficult to conclusively determine that any visualized particle must be PEG10-derived. In Pandya and Segel et al., they performed immunogold labeling of PEG10, but exhibit the same lack of negative controls. Our experiment improves on this literature by showing an absence of particles in the absence of PEG10 transfection in HEK cells. With this data, we have showed integrity of particles released by electron microscopy.

This experiment also enriches the present study by showing a profound loss of these particles in the 25% iodixanol fraction upon RLT8C co-transfection. By separating fractions, we were also able to show a more significant effect of RTL8 on the abundance of PEG10-derived VLPs by western blot in 25% iodixanol as compared to the 15% iodixanol fraction. This is summarized in Figure 2e-f and we think it considerably strengthens our overall claim that RTL8 antagonizes PEG10 VLPs.

Minor:

Line 80: define/introduce RTL8 gene. what is the function of RTL8 genes? what is the length of naturally expressed RTL8? is only the NTD domain expressed? is it cell strain or tissue dependent?

We wrote a new sentence that briefly introduces RTL8 in lines 82-83.

Line 161: ul to ml.

We have fixed this line of text (now on line 167) to read: “…methanol:chloroform precipitation and 500 µL of proteinaceous material…” (1/2 mL)

Line 364: indicate what percentage of homology/conservation supports the ambiguous statement "strong resemblance".

We thank the reviewer for pointing out the ambiguity of this statement and have improved this section. A simple alignment shows that PEG10 and RTL8C contain only ~30% identity. However, their predicted structures are nearly identical. We have enhanced this section with a new subpanel to Figure S3 in which we performed a DALI structural similarity comparison that revealed a high level of predicted structural homology between RTL8C and PEG10 gag. We have also updated the text on lines 387-392 to reflect this information. 

Line 429: when the authors say "VLP release", readers might be confused and assume that they mean VLPs released into the medium, similar to virus release without lytic phase. Please revise the wording at this point.

We have altered the text on lines 568-569 to say: “To determine whether RTL8 was inhibiting PEG10-derived VLP release from cells through protein-protein interactions…”

Line 466: This statement could be somewhat confusing, as the term "self-association" could lead the reader to think of the assembly pathway rather than the final protein network structure, so I recommend rewriting this sentence.

We have rewritten this sentence to clarify our meaning. It now reads on lines 609-610: “During retroelement capsid assembly, NTD lobes from capsid monomers assemble into penta- and hexameric cones”

Line 476: In comparing the results in Figures 3 and S5, this reviewer assumes that the IP experiments in Figure 3 are performed with conditioned medium and those in Figure S5 with lysate. Is this the case? If so, why are no tubulin bands visible in the HA-Gag and HA-NTD lanes?

The IP experiments in Figure 3e and Figure S5c were performed on cell lysate, not on conditioned medium. We do not think that Tubulin binds to PEG10 with any specificity, and so we do not expect to see it in eluates of our co-immunoprecipitations. The reason for the Tubulin contamination in the eluate of HACTD in Figure S5c is likely due to an overload of the IP leading to nonspecific pulldown of Tubulin. With that in mind, even despite this caveat, we found it notable that CTD is not pulled down with RTL8C, which is why we included it in the supplement. This caveat is mentioned in the Figure legend of S5.

Lane 538 and Figure 5c. This reviewer is aware that these experiments are somewhat noisy, so conclusions should be drawn with caution. For example, for this reviewer, Figure 5c does not corroborate that intracellular RTL8 content inversely correlates with PEG10-derived VLPs. What is the correlation coefficient of the fit shown in Figure 5c, what are the errors in these data, and can the authors show these data with corresponding error bars?

It is fair to say that RTL8 content does not clearly inversely correlate with VLP abundance, and instead we can more accurately state that we observed very low VLP abundance in a cell line with high RTL8 levels, but that RTL8 is clearly not the only regulator of VLPs as there were cell lines with low RTL8 that also have low VLP yield (Figure 3). This has been updated in the Figure 7c and in the following text.

 On lines 700-702, we have simplified our discussion of Figure 7c, saying “…We found that two of the cell lines with the lowest RTL8 expression had high VLP yields, while cells with the highest level of RTL8 protein had no detectable PEG10-derived VLPs in conditioned medium (Figure 7c).”

 On lines 745-746, we close discussion of this data by saying “…further work is needed to understand the determinants of PEG10-derived VLP release from cells.”

Line 543: This reviewer finds it confusing to normalize RTL8 levels across all cell lines. Similar to tubulin, can the authors use another band to normalize the data within each cell line? For example, would it be possible to use another vector expressing an excreted protein as a reference? This might be a better approach, as critical differences could be found depending on the cell line used. What would the ratios be if the data were normalized within each cell line? Could it be that RTL8 levels do not correlate with VLP production?

Yes, it could be that RTL8 levels do not correlate with VLP production, but could instead correlate with ALIX, Tsg101, or a number of other regulators of particle release. Unfortunately, because of time and sample constraints, we were not able to perform a new cohort of VLP harvests and western blot for other putative regulators of VLP release. We have mentioned this as a future direction in our Discussion on lines 819-821.

Reviewer #2: This is a sound study of the human retro element Gag protein expression and VLP assembly, that can be suppressed by the expression of the additional retro element coding for the N terminal part of that Gag only. The authors show convincingly that both gag and gag pol are expressed in the cell and are present in the lysates of different type cells in comparable quantities. However, the amount of the released VLPs appeared to vary strongly with the cell type. The authors have shown that the VLP release is modulated strongly by the level of NTD of Gag expression that co-assembles with Gag and GagPol, but gets incorporated into VLP in small quantities only, and supresses the VLP assembly over all. This is an important finding both for beginning to understand the mechanism of Gag assembly, as well as for the VLP cargo delivery applications.

I wonder if the RTL8 supresses Gag assembly because it lacks the RNA-binding NC domain? The retroviral cationic NC domains in the CTDs of Gag are typically supporting the VLP assembly via RNA binding. Lacking NC and RNA binding the RTL8-incorporated VLPs may be simply very unstable. Therefore, it is worth it to follow up on that study and to maybe test the retro element derived protein containing the C-terminal portion of Gag with its NC domain in its effect on VLP formation. It is possible that this protein will be much better incorporated into VLPs and not as supressing for its assembly.

We thank the reviewer for this thought-provoking point about the zinc finger promoting stability of PEG10-derived VLPs. In preliminary experiments, we found that truncation of the zinc finger of PEG10 gag (which we referred to as capsid, or CA) resulted in a decreased crude VLP yield compared to full-length gag protein (Rebuttal Figure 1 below). This finding is consistent with the idea that the zinc finger promotes stabilization of VLPs, although another fragment missing the NC domain, capsidCTD, appeared to promote relatively normal levels of VLP release. This Figure is not included in the paper because (1) we only have two biological replicates, and (2) with the recent improvement in our VLP purification method, we would want to do this using iodixanol fractionation instead of a sucrose cushion. To do this series of experiments would take considerable time beyond what was available to us for this revision. 

For these reasons, we considered further testing of RTL8 and PEG10 mutants to be beyond the scope of the current revision and of interest for future follow-up work. As mentioned below, though, we have included discussion of this possibility on lines 831-833.

Rebuttal Figure 1: Crude VLP yield of different PEG10 truncations, including two with the NC domain removed. a) Western blot of whole cell lysate from HEK293 cells transfected with different HA-tagged PEG10 constructs. Capsid = AA1-260, capsidNTD = AA1-160, capsidCTD = AA161-259, and capsidNTD + capsidCTD was a cotransfection of the two. b) Western blot of crude VLP preparation by sucrose cushion of the same constructs in HEK293 cells. c) Quantitation of results in (b) normalizing the VLP yield of all HA-positive bands to the lysate levels. n = 2 independent experiments.

The paper would improve if the authors could put forward a few hypothesis of the mechanism of supression of the VLP assembly by RTL8 in the discussion.

We welcome the opportunity to expand our discussion of the putative mechanism of RTL8 inhibition of PEG10-derived VLPs and have introduced additional speculation and hypotheses on lines 827-837 of our Discussion.

---

## [Decision Letter · Decision Letter 1]

11 Sep 2024

The gag-like gene RTL8 antagonizes PEG10-mediated virus like particles

PONE-D-24-18574R1

Dear Dr. Whiteley,

We’re pleased to inform you that your manuscript has been judged scientifically suitable for publication and will be formally accepted for publication once it meets all outstanding technical requirements.

Kind regards,

Mauricio Comas-Garcia

Academic Editor

PLOS ONE

Reviewers' comments:

Reviewer's Responses to Questions

**Comments to the Author**

1. If the authors have adequately addressed your comments raised in a previous round of review and you feel that this manuscript is now acceptable for publication, you may indicate that here to bypass the “Comments to the Author” section, enter your conflict of interest statement in the “Confidential to Editor” section, and submit your "Accept" recommendation.

Reviewer #1: All comments have been addressed

Reviewer #2: All comments have been addressed

2. Is the manuscript technically sound, and do the data support the conclusions?

Reviewer #1: Yes

Reviewer #2: Yes

3. Has the statistical analysis been performed appropriately and rigorously? 

Reviewer #1: Yes

Reviewer #2: Yes

4. Have the authors made all data underlying the findings in their manuscript fully available?

Reviewer #1: Yes

Reviewer #2: Yes

5. Is the manuscript presented in an intelligible fashion and written in standard English?

Reviewer #1: Yes

Reviewer #2: Yes

6. Review Comments to the Author

Reviewer #1: (No Response)

Reviewer #2: I believe the authors adequately answered both my and the 1st Reviewer's questions and this ms can now be published. I am looking forward to author's future study on the VLP formation of deleted Gag protein and of its mixtures with WT Gag.

7. PLOS authors have the option to publish the peer review history of their article (what does this mean?). If published, this will include your full peer review and any attached files.

Reviewer #1: No

Reviewer #2: **Yes: **Ioulia Rouzina

---

## [Editor Report · Acceptance letter]

25 Sep 2024

PONE-D-24-18574R1 

PLOS ONE

Dear Dr. Whiteley, 

I'm pleased to inform you that your manuscript has been deemed suitable for publication in PLOS ONE. Congratulations! Your manuscript is now being handed over to our production team.

Kind regards, 

on behalf of

Dr. Mauricio Comas-Garcia 

Academic Editor

PLOS ONE